# SubgDiff: A Subgraph Diffusion Model to Improve Molecular Representation Learning

**Jiying Zhang, Zijing Liu**[*]**, Yu Wang, Bin Feng, Yu Li**[*]
International Digital Economy Academy (IDEA)
{zhangjiying,liuzijing,fengbin,liyu}@idea.edu.cn
https://github.com/IDEA-XL/SubgDiff

## Abstract

Molecular representation learning has shown great success in advancing AI-based drug discovery. A key insight of many recent works is that the 3D geometric structure of molecules provides essential information about their physicochemical properties. Recently, denoising diffusion probabilistic models have achieved impressive performance in molecular 3D conformation generation. However, most existing molecular diffusion models treat each atom as an independent entity, overlooking the dependency among atoms within the substructures. This paper introduces a novel approach that enhances molecular representation learning by incorporating substructural information in the diffusion model framework. We propose a novel diffusion model termed SubgDiff for involving the molecular subgraph information in diffusion. Specifically, SubgDiff adopts three vital techniques: i) subgraph prediction, ii) expectation state, and iii) $k$-step same subgraph diffusion, to enhance the perception of molecular substructure in the denoising network. Experiments on extensive downstream tasks, especially the molecular force predictions, demonstrate the superior performance of our approach.

## 1   Introduction

Molecular representation learning (MRL) has attracted tremendous attention due to its significant role in learning from limited labeled data for applications like AI-based drug discovery [37, 3, 4, 25] and material science [33]. From the perspective of physical chemistry, the 3D molecular conformation can largely determine the properties of molecules and the activities of drugs [6, 7]. Thus, numerous geometric neural network architectures and self-supervised learning strategies have been proposed to explore 3D molecular structures to improve performance on downstream molecular property prediction tasks [35, 56, 23, 58].

Meanwhile, diffusion probabilistic models (DPMs) have shown remarkable power to generate realistic samples, especially in synthesizing high-quality images and videos [40, 10]. By modeling the generation as a reverse diffusion process, DPMs transform a random noise into a sample in the target distribution. Recently, diffusion models/flow models have also

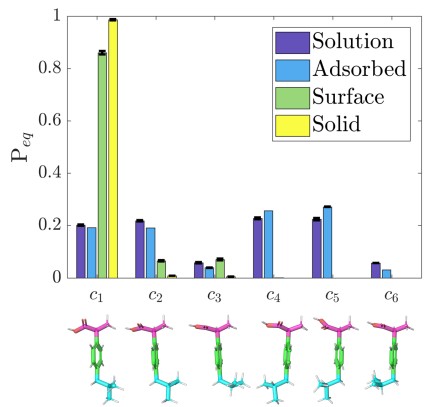

Figure 1: Equilibrium probability of the six different 3D structures (c1–c6) of the same molecule ibuprofen (C13H18O2) in four different conditions. (Adapted with permission from [27]. Copyright 2018 American Chemical Society.)

---

[*]Corresponding author

38th Conference on Neural Information Processing Systems (NeurIPS 2024).

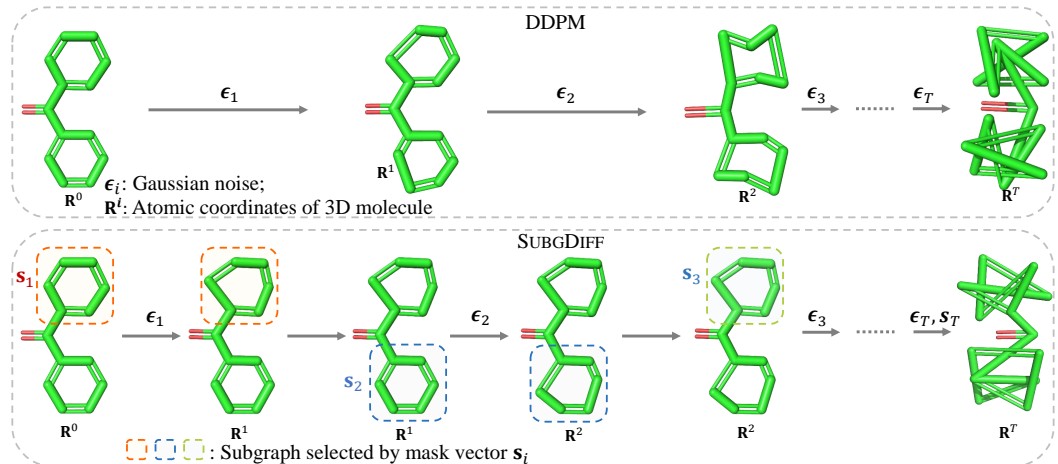

Figure 2: Comparison of forward process between DDPM [10] and subgraph diffusion. For each step, DDPM adds noise into all atomic coordinates, while subgraph diffusion selects a subset of the atoms to diffuse.

demonstrated strong capabilities of molecular 3D conformation generation [52, 16, 59, 43, 61, 59]. The training process of a DPM for conformation generation can be viewed as the reconstruction of the original conformation from a noisy version, where the noise is modulated by different time steps. Consequently, the denoising objective in the diffusion model can naturally be used as a self-supervised representation learning technique [32]. Inspired by this intuition, several works have used this technique for molecule pretraining [24, 56]. Despite considerable progress, the potential of DPMs for molecular representation learning has not been fully explored. We therefore raise the question: *Can we effectively enhance MRL with the denoising network (noise predictor) of DPM? If yes, how to achieve it?*

To answer this question, we first analyze the gap between the current DPMs and the characteristics of molecular structures. Most diffusion models on molecules propose to independently inject continuous Gaussian noise into the every node feature [11] or atomic coordinates of 3D molecular geometry [52, 56]. However, this approach treats each atom as an individual particle, overlooking the substructure within molecules, which is pivotal in molecular representation learning [55, 46, 28]. As shown in Figure 1, the substructures are mostly invariant in different 3D molecular conformations, which contains crucial information about the properties, such as the equilibrium distribution, crystallization and solubility [27]. As a result, uniformly adding same-scale Gaussian noise to all atoms makes it difficult for the denoising network to learn the joint distribution of global structure and local substructure, hindering the downstream prediction performance on molecular properties closely related to 3D conformation (e.g. most physicochemical properties). So here we try to tackle the previous question by designing a DPM involving the knowledge of substructures.

Toward this goal, we propose a novel diffusion model termed SubgDiff, adding distinct Gaussian noise to different substructures of 3D molecular conformation. Specifically, instead of adding the same Gaussian noise to every atomic coordinate, SubgDiff only adds noise to a randomly selected subgraph at each time step in the diffusion process (Figure 2). In the training phase of SubgDiff, a subgraph prediction task is integrated into the training objective, which explicitly directs the denoising network to capture substructure information from the molecules. Additionally, we propose two techniques: *expectation state diffusion* and *k-step same-subgraph diffusion*, to train the model effectively and enhance its sampling capability.

With the ability to capture the substructure information from the noisy 3D molecule, the denoising networks tend to gain more representation power. The experiments on various 2D and 3D molecular property prediction tasks demonstrate the superior performance of our approach. To summarize, our contributions are as follows: (1) we incorporate the substructure information into diffusion models to improve molecular representation learning; (2) we propose a new diffusion model SubgDiff that adopts subgraph prediction, expectation state and $k$-step same-subgraph diffusion to improve its sampling and training; (3) the proposed representation learning method achieves superior performance on various downstream tasks, especially molecular forces prediction.

## 2 Related work

**Diffusion models on graphs.** The diffusion models on graphs can be mainly divided into two categories: continuous diffusion and discrete diffusion. Continuous diffusion applies a Gaussian noise process on each node or edge [14, 30], including GeoDiff [52], EDM [11], SubDiff [54]. Meanwhile, discrete diffusion constructs the Markov chain on discrete space, including Digress [9] and GraphARM [18]. However, it remains open to exploring fusing the discrete characteristic into the continuous Gaussian on graph learning, although a closely related work has been proposed for images and cannot be used for generation [32]. Our work, SubgDiff, is the first diffusion model fusing subgraph, combining discrete characteristics and the continuous Gaussian.

**Conformation generation.** Various deep generative models have been proposed for conformation generation, including CVGAE [26], GraphDG [39], CGCF [50], ConfVAE [51], ConfGF [38] and GeoMol [8]. Recently, diffusion-based methods have shown competitive performance. Torsional Diffusion [16] raises a diffusion process on the hypertorus defined by torsion angles. However, it is not suitable as a representation learning technique due to the lack of local information (length and angle of bonds). GeoDiff [52] generates molecular conformation with a diffusion model on atomic coordinates. However, it views the atoms as separate particles, without considering the dependence between atoms from the substructure.

**SSL for molecular property prediction.** There exist several works leveraging the 3D molecular conformation to boost the representation learning, including GraphMVP [22], GeoSSL [24], the denoising pretraining approach raised by Zaidi et al. [56] and MoleculeSDE [23], etc. However, those studies have not considered the molecular substructure in the pertaining. In this paper, we concentrate on how to boost the perception of molecular substructure in the denoising networks through the diffusion model.

The discussion with more related works (e.g. MDM [32], MDSM [19] and SSSD [1]) can be found in Appendix B.1.

## 3 Preliminaries

**Notations.** We use $\mathbf{I}$ to denote the identity matrix with dimensionality implied by context, $\odot$ to represent the element product, and $\text{diag}(\mathbf{s})$ to denote the diagonal matrix with diagonal elements of the vector $\mathbf{s}$. If not specified, both $\epsilon$ and $z$ represent noise sampled from the standard Gaussian distribution $\mathcal{N}(\mathbf{0}, \mathbf{I})$. The topological molecular graph can be denoted as $\mathcal{G}(\mathcal{V}, \mathcal{E}, \mathbf{X})$ where $\mathcal{V}$ is the set of nodes, $\mathcal{E}$ is the set of edges, $\mathbf{X}$ is the node feature matrix, and its corresponding 3D Conformational Molecular Graph is represented as $G_{3D}(\mathcal{G}, \mathbf{R})$, where $\mathbf{R} = [R_1, \cdots, R_{|\mathcal{V}|}] \in \mathbb{R}^{|\mathcal{V}| \times 3}$ is the set of 3D coordinates of atoms.

**DDPM.** Denoising diffusion probabilistic models (DDPM) [10] is a typical diffusion model [40] which consists of a diffusion (aka forward) and a reverse process. In the setting of molecular conformation generation, the diffusion model adds noise on the 3D molecular coordinates $\mathbf{R}$ [52].

**Forward and reverse process.** Given the fixed variance schedule $\beta_1, \beta_2, \cdots, \beta_T$, the posterior distribution $q(R^{1:T}|R^0)$ that is fixed to a Markov chain can be written as

$$q(\mathbf{R}^{1:T}|\mathbf{R}^0) = \prod_{t=1}^{T} q(\mathbf{R}^t|\mathbf{R}^{t-1}); \quad q(\mathbf{R}^t|\mathbf{R}^{t-1}) = \mathcal{N}(\mathbf{R}^t; \sqrt{1-\beta_t}\mathbf{R}^{t-1}, \beta_t\mathbf{I}). \quad (1)$$

To simplify notation, we consider the diffusion on single atom coordinate $R_v$ and omit the subscript $v$ to get the general notion $R$ throughout the paper. Let $\alpha_t = 1 - \beta_t$, $\bar{\alpha}_t = \prod_{i=1}^{t}(1-\beta_i)$, and then the sampling of $R^t$ at any time step $t$ has the closed form: $q(R^t|R^0) = \mathcal{N}(R^t; \sqrt{\bar{\alpha}_t}R^0, (1-\bar{\alpha}_t)\mathbf{I})$.

The reverse process of DDPM is defined as a Markov chain starting from a Gaussian distribution $p(R^T) = \mathcal{N}(R^T; \mathbf{0}, \mathbf{I})$:

$$p_\theta(R_{0:T}) = p(R^T) \prod_{t=1}^{T} p_\theta(R^{t-1}|R^t); \quad p_\theta(R^{t-1}|R^t) = \mathcal{N}(R^{t-1}; \mu_\theta(R^t, t), \sigma_t), \quad (2)$$

where $\sigma_t = \frac{1-\bar{\alpha}_{t-1}}{1-\bar{\alpha}_t}\beta_t$ denote time-dependent constant. In DDPM, $\mu_\theta(R^t, t)$ is parameterized as $\mu_\theta(R^t, t) = \frac{1}{\alpha_t}(R^t - \frac{\beta_t}{\sqrt{1-\bar{\alpha}_t}}\epsilon_\theta(R^t, t))$ and $\epsilon_\theta$, i.e., the *denoising network*, is parameterized by a neural network where the inputs are $R^t$ and time step $t$.

**Training and sampling.** The training objective of DDPM is:

$$\mathcal{L}_{simple}(\theta) = \mathbb{E}_{t,R^0,\epsilon}[\|\epsilon - \epsilon_\theta(\sqrt{\bar{\alpha}_t}R^0 + \sqrt{1-\bar{\alpha}_t}\epsilon, t)\|^2]. \tag{3}$$

After training, samples are generated through the reverse process $p_\theta(R^{0:T})$. Specifically, $R^T$ is first sampled from $\mathcal{N}(\mathbf{0}, \mathbf{I})$, and $R^t$ in each step is predicted as follows,

$$R^{t-1} = \frac{1}{\sqrt{\alpha_t}}(R^t - \frac{1-\alpha_t}{\sqrt{1-\bar{\alpha}_t}}\epsilon_\theta(R^t, t)) + \sigma_t z, \quad z \sim \mathcal{N}(\mathbf{0}, \mathbf{I}). \tag{4}$$

## 4 SubgDiff

Directly using DDPM on atomic coordinates of 3D molecules means each atom is viewed as an independent single data point. However, the substructures play an important role in molecular generation [15] and representation learning [57]. Ignoring the inherent interactions among atoms within substructures may hinder the denoising network from learning features for molecular properties related to 3D conformation. In this paper, we propose to involve a mask operation in each diffusion step, leading to a new diffusion SubgDiff for molecular representation learning. Each mask corresponds to a subgraph in the molecular graph, aligning with the substructure in the 3D molecule. Furthermore, we incorporate a subgraph predictor and reset the state of the Markov Chain to the expectation of atomic coordinates, thereby enhancing the effectiveness of SubgDiff in sampling. Additionally, we also propose $k$-step same-subgraph diffusion for training to effectively capture the substructure information.

### 4.1 Involving subgraph into diffusion process

In the forward process of DDPM, we have $R_v^t = \sqrt{1-\beta_t}R_v^{t-1} + \sqrt{\beta_t}\epsilon_{t-1}, \forall v \in \mathcal{V}$, in which the Gaussian noise $\epsilon_{t-1}$ is injected to every atom. Moreover, the training objective in Equation 3 shows that the denoising networks would always predict a Gaussian noise for all atoms. Neither the forward nor reverse process of DDPM takes into account the substructure of the molecule. Instead, in SubgDiff, a mask vector $\mathbf{s}_t = [s_{t_1}, \cdots, s_{t_{|\mathcal{V}|}}]^\top \in \{0,1\}^{|\mathcal{V}|}$ is introduced to determine which atoms will be added noise at step $t$. The mask vector $\mathbf{s}_t$ is sampled from a discrete distribution $p_{\mathbf{s}_t}(\mathcal{S} \mid \mathcal{G})$ to select a subset of the atoms. In molecular graphs, the discrete

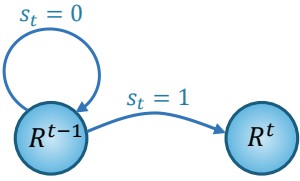

Figure 3: The Markov Chain of SubgDiff is a lazy Markov Chain.

mask distribution $p_{\mathbf{s}_t}(\mathcal{S} \mid \mathcal{G})$ is equivalent to the subgraph distribution, defined over a predefined sample space $\chi = \{G_{\text{sub}}^i\}_{i=1}^N$, where each sample is a connected subgraph extracted from $G$. Further, the distribution $p_{\mathbf{s}_t}(\mathcal{S} \mid \mathcal{G})$ should keep the selected connected subgraph to cohere with the molecular substructures. Here, we adopt a Torsional-based decomposition method [16] (Details in Appendix A.2). With the mask vector as latent variables $\mathbf{s}_{1:t}$, the state transition of the forward process can be formulated as (Figure 3):

$$R_v^t = \begin{cases} \sqrt{1-\beta_t}R_v^{t-1} + \sqrt{\beta_t}\epsilon_{t-1} & \text{if } s_{t_v} = 1 \\ R_v^t & \text{if } s_{t_v} = 0, \end{cases} \tag{5}$$

which can be rewritten as $R_v^t = \sqrt{1-s_{t_v}\beta_t}R_v^{t-1} + \sqrt{s_{t_v}\beta_t}\epsilon_{t-1}$.

The posterior distribution $q(R^{1:T}|R^0, s_{1:T})$ can be expressed as matrix form:

$$q(\mathbf{R}^{1:T}|\mathbf{R}^0, \mathbf{s}_{1:T}) = \prod_{t=1}^T q(\mathbf{R}^t|\mathbf{R}^{t-1}, \mathbf{s}_t); q(\mathbf{R}^t|\mathbf{R}^{t-1}, \mathbf{s}_t) = \mathcal{N}(\mathbf{R}^t; \sqrt{1-\beta_t\text{diag}(\mathbf{s}_t)}\mathbf{R}^{t-1}, \beta_t\text{diag}(\mathbf{s}_t)\mathbf{I}). \tag{6}$$

To simplify the notation, we consider the diffusion on a single node $v$ and omit the subscript $v$ in $R_v^t$ and $s_{t_v}$ to get the notion $R^t$ and $s_t$. By defining $\gamma_t := 1 - s_t\beta_t, \bar{\gamma}_t := \prod_{i=1}^t(1 - s_t\beta_t)$, the closed form of sampling $R^t$ given $R^0$ is

$$q(R^t|R^0, s_{1:t}) = \mathcal{N}(R^t; \sqrt{\bar{\gamma}_t}R^0, (1-\bar{\gamma}_t)\mathbf{I}). \tag{7}$$

### 4.2 Reverse process learning

The reverse process is decomposed as follows:

$$p_{\theta,\vartheta}(R^{0:T}, s_{1:T}) = p(R^T) \prod_{t=1}^{T} p_\theta(R^{t-1}|R^t, s_t) p_\vartheta(s_t|R^t), \tag{8}$$

where $p_\theta(R^{t-1}|R^t, s_t)$ and $p_\theta(s_t|R^t)$ are both learnable models. In the context of molecular learning, the model can be regarded as first predicting which subgraph $\mathbf{s}_t$ should be denoised and then using the noise prediction network $p_\theta(R^{t-1}|R^t, s_t)$ to denoise the node position in the subgraph.

However, it is tricky to generate a 3D structure by adopting the typical training and sampling method used in Ho et al. [10]. Specifically, following Ho et al. [10], the reverse process can be optimized by maximizing the variational lower bound (VLB) of $\log p(R^0)$ as follows,

$$\log p(R^0) \geq \sum_{t=1}^{T} \underbrace{\mathbb{E}_{q(R^t, s_t|R^0)} \left[ \log \frac{p_\vartheta(s_t|R^t)}{q(s_t)} \right]}_{\text{subgraph prediction term}} + \underbrace{\mathbb{E}_{q(R^1, s_1|R^0)} \left[ \log p_\theta(R^0|R^1) \right]}_{\text{reconstruction term}} -$$

$$\underbrace{\mathbb{E}_{q(s_{1:t})} D_{\mathrm{KL}}(q(R^T|R^0, s_{1:T}) \parallel p(R^T))}_{\text{prior matching term}} - \sum_{t=2}^{T} \underbrace{\mathbb{E}_{q(R^t, s_{1:t}|R^0)} \left[ D_{\mathrm{KL}}(q(R^{t-1}|R^t, R^0, s_{1:t}) \parallel p_\theta(R^{t-1}|R^t, s_t)) \right]}_{\text{denoising matching term}}. \tag{9}$$

Details of the derivation are provided in the Appendix D.2. The subgraph predictor $p_\vartheta(s_t|R^t)$ in the first term can be parameterized by a node classifier $s_\vartheta$. For the denoising matching term that is closely related to sampling, by Bayes rule, the posterior $q(R^{t-1}|R^t, R^0, s_{1:t})$ can be written as:

$$q(R^{t-1}|R^t, R^0, s_{1:t}) \propto \mathcal{N}(R^{t-1}; \mu_q(R^t, R^0, s_{1:t}, \epsilon_0), \sigma_q^2(t)), \tag{10}$$

$$\mu_q(R^t, R^0, s_{1:t}, \epsilon_0) = \frac{1}{\sqrt{1 - \beta_t s_t}} (R^t - \frac{\beta_t s_t}{\sqrt{(1 - \beta_t s_t)(1 - \bar{\gamma}_{t-1}) + \beta_t s_t}} \epsilon_0),$$

where $\sigma_q(t)$ is the standard deviation and $s_{1:t-1}$ are contained in $\bar{\gamma}_{t-1}$. Following DDPM and parameterizing $p_\theta(R^{t-1}|R^t, s_t)$ as $\mathcal{N}(R^{t-1}; \mu_q(R^t, R^0, s_{1:t-1}, s_\vartheta(\mathcal{G}, R^t, t), \epsilon_\theta(\mathcal{G}, R^t, t)), \sigma_q(t)\mathbf{I})$, the training objective is

$$\mathcal{L}_{simple}(\theta, \vartheta) = \mathbb{E}_{t, \mathbf{R}^0, \mathbf{s}_t, \epsilon}[\|\mathrm{diag}(\mathbf{s}_t)(\epsilon - \epsilon_\theta(\mathcal{G}, \mathbf{R}^t, t))\|^2 + \lambda \mathrm{BCE}(\mathbf{s}_t, \mathbf{s}_\vartheta(\mathcal{G}, \mathbf{R}^t, t))], \tag{11}$$

where $\mathrm{BCE}(\mathbf{s}_t, \mathbf{s}_\vartheta)$ is the binary cross entropy loss, $\lambda$ is the weight used for the trade-off, and $\mathbf{s}_\vartheta$ is the subgraph predictor implemented as a node classifier with $G_{3D}(\mathcal{G}, \mathbf{R}^t)$ as input and shares a molecule encoder with $\epsilon_\theta$. The BCE loss employed here uses the subgraph selected at time-step $t$ as the target, thereby explicitly compelling the denoising network to capture substructure information from molecules. Eventually, the $s_\vartheta$ can be used to infer the mask vector $\hat{\mathbf{s}}_t = s_\vartheta(\mathcal{G}, \mathbf{R}^t, t)$ during sampling. Thus, the sampling process is:

$$R^{t-1} = \mu_q(R^t, R^0, s_{1:t-1}, s_\vartheta(\mathcal{G}, R^t, t), \epsilon_\theta(\mathcal{G}, R^t, t)) + \sigma_q(t)z. \tag{12}$$

However, using Equation 11 and Equation 12 directly for training and sampling faces two issues. First, the inability to access $s_{1:t-1}$ in $\mu_q$ during the sampling process hinders the step-wise denoising procedure, posing a challenge to the utilization of conventional sampling methods in this context. Inferring $s_{1:t-1}$ solely from $R^t$ using another model $p_\theta(s_{1:t-1}|R^t, s_t)$ is also difficult due to the intricate modulation of noise introduced in $R^t$ through multi-step Gaussian noising. Second, training the subgraph predictor with Equation 11 is challenging. To be specific, the subgraph predictor should be capable of perceiving the sensible noise change between time steps $t - 1$ and $t$. However, the noise scale $\beta_t$ is relatively small when $t$ is small, especially if the diffusion step is large (e.g. 1000). As a result, it is difficult to precisely predict the subgraph.

Next, to effectively tackle the above issues, we design two techniques: *expectation state diffusion* and *k-step same subgraph diffusion*.

### 4.3 Expectation state diffusion.

We first devise a new way to calculate the denoising term. To eliminate the effect of mask series $s_{1:t}$ and improve the training of subgraph prediction loss, we use a new lower bound of the denoising term as follows:

$$\mathbb{E}_{q(R^t, s_{1:t}|R^0)} \left[ D_{\mathrm{KL}}(q(R^{t-1}|R^t, R^0, s_{1:t}) \parallel p_\theta(R^{t-1}|R^t, s_t)) \right]$$
$$\leq \mathbb{E}_{q(R^t, s_{1:t}|R^0)} \left[ D_{\mathrm{KL}}(\hat{q}(R^{t-1}|R^t, R^0, s_{1:t}) \parallel p_\theta(R^{t-1}|R^t, s_t)) \right],$$

where $\hat{q}(R^{t-1}|R^t, R^0, s_{1:t})$ is defined as $\frac{q(R^t|\mathbb{E}_{s_{1:t-1}}R^{t-1}, R^0, s_t)q(\mathbb{E}_{s_{1:t-1}}R^{t-1}|R^0)}{q(R^t|R^0, s_{1:t})}$. It is an approximated posterior that only relies on the *expectation* of $R_t$ and $s_t$. This lower bound defines a new forward process, in which state 0 to state $t-1$ use the $\mathbb{E}_{s_{1:t-1}}\mathbf{R}^{t-1}$ and state $t$ remains as Equation 6. Assume each node $v \in \mathcal{V}$, $s_{t_v} \sim Bern(p)$ (i.i.d. w.r.t. $t$). Formally, we have

$$q(R^t|R^{t-1}, s_t) = \mathcal{N}(R^t; \sqrt{1 - s_t\beta_t}\mathbb{E}R^{t-1}, (s_t\beta_t)\mathbf{I}), \tag{13}$$

$$q(\mathbb{E}R^{t-1}|R^0, s_{1:t-1}) = \mathcal{N}(\mathbb{E}R^{t-1}; \prod_{i=1}^{t-1}\sqrt{\alpha_i}R^0, p^2\sum_{i=1}^{t-1}\prod_{j=i+1}^{t-1}\alpha_j\beta_i I), \tag{14}$$

where $\alpha_i := (p\sqrt{1 - \beta_i} + 1 - p)^2$ and $\bar{\alpha}_t := \prod_{i=1}^{t}\alpha_i$ are general form of $\alpha_j$ and $\bar{\alpha}_j$ in DDPM ($p = 1$), respectively. Intuitively, this process is equivalent to using mean state $\mathbb{E}R^{t-1}$ to replace $R^{t-1}$ during the forward process. This estimation is reasonable since the expectation $\mathbb{E}_{s_{1:t-1}}R^{t-1}$ is like a cluster center of $R^{t-1}$, which can represent $R^{t-1}$ properly. Thus, the approximated posterior becomes

$$\hat{q}(R^{t-1}|R^t, R^0, s_{1:t}) \propto \mathcal{N}(R^{t-1}; \mu_{\hat{q}}(R^t, R^0, s_{1:t}, \epsilon_0), \sigma_{\hat{q}}^2(t)), \tag{15}$$

where

$$\mu_{\hat{q}}(R^t, R^0, s_{1:t}, \epsilon_0) := \frac{1}{\sqrt{1 - \beta_t s_t}}(R^t - \frac{s_t\beta_t}{\sqrt{(s_t\beta_t + (1 - s_t\beta_t)p^2\sum_{i=1}^{t-1}\frac{\bar{\alpha}_{t-1}}{\bar{\alpha}_i}\beta_i}}\epsilon_0)$$

$$\sigma_{\hat{q}}^2(t) := s_t\beta_t p^2 \sum_{i=1}^{t-1}\frac{\bar{\alpha}_{t-1}}{\bar{\alpha}_i}\beta_i/(s_t\beta_t + p^2(1 - s_t\beta_t)\sum_{i=1}^{t-1}\frac{\bar{\alpha}_{t-1}}{\bar{\alpha}_i}\beta_i).$$

We parameterize $p_\theta(R^{t-1}|R^t, s_t)$ as $\mathcal{N}(R^{t-1}; \mu_{\hat{q}}(R^t, R^0, s_{1:t-1}, s_\vartheta(\mathcal{G}, R^t, t), \epsilon_\theta(R^t, t)), \sigma_{\hat{q}}(t)\mathbf{I})$, and adopt the same training objective as Equation 11. By employing the sampling method $R^{t-1} = \mu_{\hat{q}}(R^t, R^0, s_{1:t-1}, s_\vartheta(\mathcal{G}, R^t, t), \epsilon_\theta(R^t, t)) + \sigma_{\hat{q}}(t)z$, we observe that the proposed expectation state enables a step-by-step execution of the sampling process. Moreover, using expectation is beneficial to reduce the complexity of $R^t$ for predicting the mask $s_t$ during training. This will improve the denoising network to perceive the substructure when we use the diffusion model for self-supervised learning.

## 4.4 $k$-step same-subgraph diffusion.

To reduce the complexity of the mask series $(\mathbf{s}_1, \mathbf{s}_2, \cdots, \mathbf{s}_T)$ and accumulate more noise on the same subgraph for facilitating the convergence of the subgraph prediction loss, we generalize the one-step subgraph diffusion to $k$-step same subgraph diffusion (Figure 8 in Appendix), in which the selected subgraph will be continuously diffused $k$ steps. After that, the difference between the selected and unselected parts will be distinct enough to help the subgraph predictor perceive it. The forward process of $k$-step same subgraph diffusion can be written as ($t > k, k \in \mathbb{N}$):

$$q(R^t|R^{t-k}, s_{t-k+1:t}) = \mathcal{N}\left(R^t, \sqrt{\prod_{i=t-k+1}^{t}(1 - s_{t-k+1}\beta_i)}R^{t-k}, \sigma_t^k\right), \tag{16}$$

where $\sigma_t^k = (1 - \prod_{i=t-k}^{t}(1 - s_{t-k+1}\beta_i))\mathbf{I}$.

## 4.5 Training and sampling of SubgDiff

By combining the expectation state and $k$-step same-subgraph diffusion, SubgDiff first divides the entire diffusion step $T$ into $T/k$ diffusion intervals. In each interval $[ki, k(i + 1)]$, the mask vectors $\{\mathbf{s}_j\}_{j=ki+2}^{k(i+1)}$ are equal to $\mathbf{s}_{ki+1}$. SubgDiff then adopts the expectation state at the split time step $\{ik|i = 1, 2, \cdots\}$ to eliminate the effect of $\{\mathbf{s}_{ik+1}|i = 1, 2, \ldots\}$, that is, gets the expectation of $\mathbb{E}\mathbf{R}^{ik}$ at step $ik$ w.r.t. $\mathbf{s}_{ik+1}$. Overall, the diffusion process of SubgDiff is a two-phase diffusion process. In the first phase, the state 1 to state $k\lfloor(t - 1)/k\rfloor$ use the expectation state diffusion, while in the second phase, state $k(\lfloor(t - 1)/k\rfloor) + 1$ to state $t$ use the $k$-step same subgraph diffusion (see Figure 4). With $m := \lfloor(t - 1)/k\rfloor$, the two phases can be formulated as follows,

**Phase I**: Step $0 \to km$: $\mathbb{E}_{s_{1:km}}R^{km} = \sqrt{\bar{\alpha}_m}R^0 + p\sqrt{\sum_{l=1}^{m}\frac{\bar{\alpha}_m}{\bar{\alpha}_l}(1 - \prod_{i=(l-1)k+1}^{kl}(1 - \beta_i))}\epsilon_0$,

where $\alpha_j = (p\sqrt{\prod_{i=(j-1)k+1}^{kj}(1 - \beta_i)} + 1 - p)^2$ is a general forms of $\alpha_j$ in Equation 14 (in which

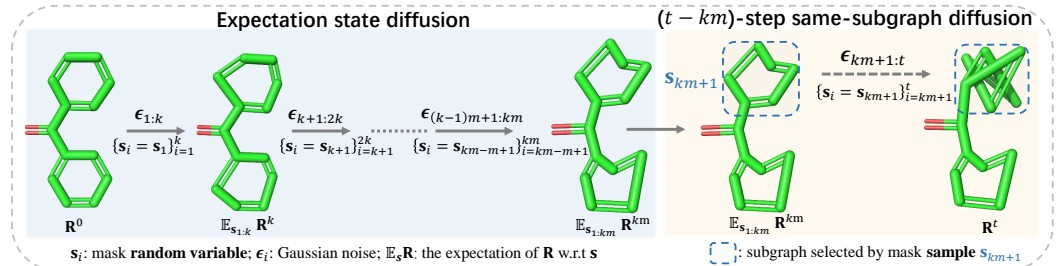

Figure 4: The forward process of SubgDiff. The state 0 to $km$ uses the expectation state and the mask variables are the same in the interval $[ki, ki + k], i = 0, 1, ..., m - 1$. The state $km + 1$ to $t$ applies the same subgraph diffusion.

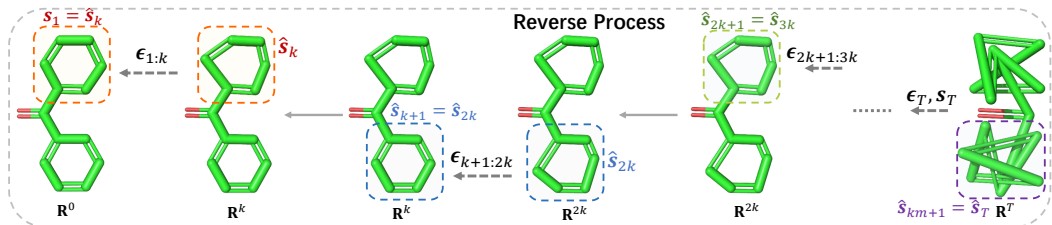

Figure 5: The reverse process of SubgDiff. The selected subgraph $\mathbf{s}$ is the same in the interval $[ki, min(ki + k, T)]$, $i = 0, ..., m$

case $k = 1$) and $\bar{\alpha}_i = \prod_{j=1}^{t} \alpha_j$. In the rest of the paper, $\alpha_j$ denotes the general version without a special statement. Actually, $\mathbb{E}_{s_{1:km}} R^{km}$ only calculate the expectation w.r.t. random variables $\{\mathbf{s}_{ik+1} | i = 1, 2, \cdots \}$.

**Phase II**: Step $km + 1 \rightarrow t$: The phase is a $(t - km)$-step same mask diffusion. $R^t = \sqrt{\prod_{i=km+1}^{t}(1 - \beta_i s_{km+1})} \mathbb{E}_{s_{1:km}} R^{km} + \sqrt{1 - \prod_{i=km+1}^{t}(1 - \beta_i s_{km+1})} \epsilon_{km}$.

Let $\gamma_i = 1 - \beta_i s_{km+1}$, $\bar{\gamma}_t = \prod_{i=1}^{t} \gamma_i$, and $\bar{\beta}_t = \prod_{i=1}^{t}(1 - \beta_i)$. We can drive the single-step state transition: $q(R^t | R^{t-1}) = \mathcal{N}(R^t; \sqrt{\gamma_t} R^{t-1}, (1 - \gamma_t)\mathbf{I})$ and

$$q(R^{t-1} | R^0) = \mathcal{N}(R^{t-1}; \sqrt{\frac{\bar{\gamma}_{t-1}\bar{\alpha}_m}{\bar{\gamma}_{km}}} R^0, \delta I); \quad \delta := \frac{\bar{\gamma}_{t-1}}{\bar{\gamma}_{km}} p^2 \sum_{l=1}^{m} \frac{\bar{\alpha}_m}{\bar{\alpha}_l}(1 - \frac{\bar{\beta}_{kl}}{\bar{\beta}_{(l-1)k}}) + 1 - \frac{\bar{\gamma}_{t-1}}{\bar{\gamma}_{km}}. \quad (17)$$

Then we reuse the training objective in Equation 11 as the objective of SubgDiff:

$$\mathcal{L}_{simple}(\theta, \vartheta) = \mathbb{E}_{t, \mathbf{R}^0, \mathbf{s}_t, \epsilon}[\|\text{diag}(\mathbf{s}_t)(\epsilon - \epsilon_\theta(\mathcal{G}, \mathbf{R}^t, t))\|^2 + \lambda \text{BCE}(\mathbf{s}_t, s_\vartheta(\mathcal{G}, \mathbf{R}^t, t))], \quad (18)$$

where $\mathbf{R}^t$ is calculated by Equation 17.

**Sampling.** Although the forward process uses the expectation state w.r.t. $\mathbf{s}$, we can only update the mask $\hat{\mathbf{s}}_t$ at $t = ik, i = 1, 2, \cdots$ because sampling only needs to get a subgraph from the distribution in the $k$-step interval. Eventually, adopting the $\delta$ defined in Equation 17, the sampling process (Figure 5) is shown below,

$$R^{t-1} = \frac{1}{\sqrt{\gamma_t}}(R^t - \frac{\hat{s}_{km+1}\beta_t}{\sqrt{\gamma_t\delta + \hat{s}_{km+1}\beta_t}}\epsilon_\theta(R^t, t)) + \frac{\sqrt{\hat{s}_{km+1}\beta_t}\sqrt{\frac{\bar{\gamma}_{t-1}}{\bar{\gamma}_{km}}p^2\sum_{l=1}^{m}\frac{\bar{\alpha}_m}{\bar{\alpha}_l}(1 - \frac{\bar{\beta}_{kl}}{\bar{\beta}_{(l-1)k}}) + 1 - \frac{\bar{\gamma}_{t-1}}{\bar{\gamma}_{km}}}}{\sqrt{\gamma_t\delta + \hat{s}_{km+1}\beta_t}} z, \quad (19)$$

where $z \sim \mathcal{N}(\mathbf{0}, \mathbf{I})$, $m = \lfloor (t - 1)/k \rfloor$ and $\hat{s}_{km+1} = s_\vartheta(\mathcal{G}, R^{km+k}, km + k)$. The subgraph selected by $\hat{s}_{km+1}$ will be generated in from the steps $km + k$ to $km$. The mask predictor can be viewed as a discriminator of important subgraphs, indicating the optimal subgraph should be recovered in the next $k$ steps. After one subgraph (substructure) is generated properly, the model can gently fine-tune the other parts of the molecule (c.f. the video in supplementary material). This subgraph diffusion would intuitively increase the robustness and generalization of the generation process, which is also verified by the experiments in Section 5.2. The training and sampling algorithms of SubgDiff are summarized in Algorithm 1 and Algorithm 2.

**Algorithm 1:** Training SubgDiff

---

**Input:** A molecular graph $G_{3D}$, $k$ for same mask diffusion, $m := \lfloor (t-1)/k \rfloor$

Sample $t \sim \mathcal{U}(1, ..., T)$, $\epsilon \sim \mathcal{N}(\mathbf{0}, \mathbf{I})$

Sample $\mathbf{s}_{km+1} \sim p_{s_{km+1}}(\mathcal{S} \mid \mathcal{G})$        ▷ Sample a subgraph

$\mathbf{R}^t \leftarrow q(\mathbf{R}^t | \mathbf{R}^0)$        ▷ Equation 17

$\mathcal{L}_1 = \mathrm{BCE}(\mathbf{s}_{km+1}, s_\vartheta(\mathcal{G}, \mathbf{R}^t, t))$        ▷ Subgraph prediction loss

$\mathcal{L}_2 = \|\mathrm{diag}(\mathbf{s}_{km+1})(\epsilon - \epsilon_\theta(\mathcal{G}, \mathbf{R}^t, t))\|^2$        ▷ Denoising loss

$\mathrm{optimizer.step}(\mathbb{E}_{t, \mathbf{R}^0, \mathbf{s}_t, \epsilon}[\lambda \mathcal{L}_1 + \mathcal{L}_2])$        ▷ Optimize parameters $\theta, \vartheta$

---

**Algorithm 2:** Sampling from SubgDiff

---

$k$ is the same as training, for $k$-step same-subgraph diffusion;

Sample $\mathbf{R}^T \sim \mathcal{N}(\mathbf{0}, \mathbf{I})$        ▷ Random noise initialization

**for** $t$ = $T$ **to** $1$ **do**
    $\mathbf{z} \sim \mathcal{N}(\mathbf{0}, \mathbf{I})$ if $t > 1$, else $\mathbf{z} = \mathbf{0}$        ▷ Random noise
    **If** $t\%k == 0$ or $t == T$: $\hat{\mathbf{s}} \leftarrow s_\vartheta(\mathcal{G}, \mathbf{R}^t, t)$        ▷ Subgraph prediction
    $\hat{\epsilon} \leftarrow \epsilon_\theta(\mathcal{G}, \mathbf{R}^t, t)$        ▷ Posterior
    $\mathbf{R}^{t-1} \leftarrow$ Equation 19        ▷ sampling
**end**

**return** $\mathbf{R}^0$

---

## 5 Experiments

We conduct experiments to address the following two questions: 1) Can substructures improve the representation ability of the denoising network when using diffusion as self-supervised learning? 2) How does the proposed subgraph diffusion affect the generative ability of the diffusion models? For the first question, we employ SubgDiff as a denoising pretraining task and evaluate the performances of the denoising network on various downstream tasks. For the second one, we compare SubgDiff with the vanilla diffusion model GeoDiff [52] on the task of molecular conformation generation.

### 5.1 SubgDiff improves molecular representation learning

To verify the introduced substructure in the diffusion can enhance the denoising network for representation learning, we pretrain with SubgDiff objective and finetune on various downstream tasks.

**Dataset and settings.** For pretraining, we follow [23] and use PCQM4Mv2 dataset [12]. It's a sub-dataset of PubChemQC [29] with 3.4 million molecules with 3D geometric conformations. We use various molecular property prediction datasets as downstream tasks. For tasks with 3D conformations, we consider the dataset MD17 and follow the literature [35, 36, 24] of using 1K for training and 1K for validation, while the test set (from 48K to 991K) is much larger. For downstream tasks with only 2D molecule graphs, we use eight molecular property prediction tasks from MoleculeNet [48].

**Pretraining framework.** To explore the potential of the proposed method for representation learning, we consider MoleculeSDE [23], a SOTA pretraining framework, to be the training backbone, where SubgDiff is used for the $2D \rightarrow 3D$ model and the mask operation is extended to the node feature and graph adjacency for the $3D \rightarrow 2D$ model. The details can be found in Appendix A.4.2.

**Baselines.** For 3D tasks, we incorporate two self-supervised methods [Type Prediction, Angle Prediction], and three contrastive methods [InfoNCE [31] and EBM-NCE [22] and 3D InfoGraph [24]]. Two denoising baselines are also included [GeoSSL [24], Denoising [56] and MoleculeSDE]. For 2D tasks, the baselines are AttrMask, ContexPred [13], InfoGraph [45], MolCLR [47], 3D InfoMax [44], GraphMVP [22] and MoleculeSDE.

**Results.** As shown in Table 1 and Table 2, SubgDiff outperforms MoleculeSDE in both 2D and 3D downstream tasks. Particularly, the significant improvement on the MD17 force prediction task, which is closely related to 3D molecular conformation, demonstrates that the introduced subgraph diffusion helps the perception of molecular 3D structure in the denoising network during pretraining. Further, SubgDiff achieves SOTA performance compared to all the baselines. This also indicates that the proposed SubgDiff objective is promising for molecular representation learning due to the

Table 1: Results (mean absolute error) on MD17 **force** prediction. The best and second best results are marked in bold and underlined.

| Pretraining | Aspirin ↓ | Benzene ↓ | Ethanol ↓ | Malonaldehyde ↓ | Naphthalene ↓ | Salicylic ↓ | Toluene ↓ | Uracil ↓ |
|---|---|---|---|---|---|---|---|---|
| – (random init) | 1.203 | 0.380 | 0.386 | 0.794 | 0.587 | 0.826 | 0.568 | 0.773 |
| Type Prediction | 1.383 | 0.402 | 0.450 | 0.879 | 0.622 | 1.028 | 0.662 | 0.840 |
| Angle Prediction | 1.542 | 0.447 | 0.669 | 1.022 | 0.680 | 1.032 | 0.623 | 0.768 |
| 3D InfoGraph | 1.610 | 0.415 | 0.560 | 0.900 | 0.788 | 1.278 | 0.768 | 1.110 |
| InfoNCE | 1.132 | 0.395 | 0.466 | 0.888 | 0.542 | 0.831 | 0.554 | 0.664 |
| EBM-NCE | 1.251 | 0.373 | 0.457 | 0.829 | 0.512 | 0.990 | 0.560 | 0.742 |
| Denoising | 1.364 | 0.391 | 0.432 | 0.830 | 0.599 | 0.817 | 0.628 | 0.607 |
| GeoSSL | 1.107 | 0.360 | 0.357 | 0.737 | 0.568 | 0.902 | 0.484 | 0.502 |
| MoleculeSDE (VE) | 1.112 | 0.304 | 0.282 | 0.520 | 0.455 | 0.725 | 0.515 | 0.447 |
| MoleculeSDE (VP) | 1.244 | 0.315 | 0.338 | 0.488 | 0.432 | 0.712 | 0.478 | 0.468 |
| Ours | **0.880** | **0.252** | **0.258** | **0.459** | **0.325** | **0.572** | **0.362** | **0.420** |

Table 2: Results for MoleculeNet (with 2D topology only). We report the mean (and standard deviation) ROC-AUC of three random seeds with scaffold splitting for each task. The backbone is GIN. The best and second best results are marked bold and underlined, respectively.

| Pre-training | BBBP ↑ | Tox21 ↑ | ToxCast ↑ | Sider ↑ | ClinTox ↑ | MUV ↑ | HIV ↑ | Bace ↑ | Avg ↑ |
|---|---|---|---|---|---|---|---|---|---|
| – (random init) | 68.1±0.59 | 75.3±0.22 | 62.1±0.19 | 57.0±1.33 | 83.7±2.93 | 74.6±2.35 | 75.2±0.70 | 76.7±2.51 | 71.60 |
| AttrMask | 65.0±2.36 | 74.8±0.25 | 62.9±0.11 | 61.2±0.12 | 87.7±1.19 | 73.4±2.02 | 76.8±0.53 | 79.7±0.33 | 72.68 |
| ContextPred | 65.7±0.62 | 74.2±0.06 | 62.5±0.31 | 62.2±0.59 | 77.2±0.88 | 75.3±1.57 | 77.1±0.86 | 76.0±2.08 | 71.28 |
| InfoGraph | 67.5±0.11 | 73.2±0.43 | 63.7±0.50 | 59.9±0.30 | 76.5±1.07 | 74.1±0.74 | 75.1±0.99 | 77.8±0.88 | 70.96 |
| MolCLR | 66.6±1.89 | 73.0±0.16 | 62.9±0.38 | 57.5±1.77 | 86.1±0.95 | 72.5±2.38 | 76.2±1.51 | 71.5±3.17 | 70.79 |
| 3D InfoMax | 68.3±1.12 | 76.1±0.18 | 64.8±0.25 | 60.6±0.78 | 79.9±3.49 | 74.4±2.45 | 75.9±0.59 | 79.7±1.54 | 72.47 |
| GraphMVP | 69.4±0.21 | 76.2±0.38 | 64.5±0.20 | 60.5±0.25 | 86.5±1.70 | 76.2±2.28 | 76.2±0.81 | 79.8±0.74 | 73.66 |
| MoleculeSDE(VE) | 68.3±0.25 | 76.9±0.23 | 64.7±0.06 | 60.2±0.29 | 80.8±2.53 | 76.8±1.71 | 77.0±1.68 | 79.9±1.76 | 73.15 |
| MoleculeSDE(VP) | 70.1±1.35 | 77.0±0.12 | 64.0±0.07 | 60.8±1.04 | 82.6±3.64 | 76.6±3.25 | 77.3±1.31 | 81.4±0.66 | 73.73 |
| Ours | **70.2±2.23** | **77.2±0.39** | **65.0±0.48** | **62.2±0.97** | **88.2±1.57** | **77.3±1.17** | **77.6±0.51** | **82.1±0.96** | **74.85** |

involvement of the knowledge of substructures during training. More results on the QM9 dataset [35] on the quantum mechanics property prediction can be found in Appendix A.4.3.

## 5.2 SubgDiff benefits conformation generation

We have proposed a new diffusion model to enhance molecular representing learning, where the base diffusion model (GeoDiff) is initially designed for conformation generation. To evaluate the effects of SubgDiff on the generative ability of diffusion models, we assess its generation performance and generalization ability. Following prior works [52], we utilize the GEOM-QM9 [34] and GEOM-Drugs [2] datasets. The detailed description of the dataset splitting, metrics and model architecture can be found in Appendix A.5.

**Conformation generation.** The comparison with GeoDiff on the GEOM-QM9 dataset is reported in Table 3. From the results, it is easy to see that SubgDiff significantly outperforms the GeoDiff baseline on both metrics (COV-R and MAT-R) across different sampling steps. It indicates that by training with the substructure information, SubgDiff has a positive effect on the conformation generation task. Moreover, SubgDiff with 500 steps achieves much better performance than GeoDiff with 5000 steps on 5 out of 8 metrics, which implies our method can accelerate the sampling efficiency (10x).

Table 3: Results for conformation generation on **GEOM-QM9** dataset with different diffusion timesteps. DDPM [10] is the sampling method used in GeoDiff. Our proposed sampling method (Algorithm 2) can be viewed as a DDPM variant. ▨ / ▨ denotes SubgDiff outperforms/underperforms GeoDiff.

| Models | Timesteps | Sampling method | COV-R (%) ↑ | | MAT-R (Å) ↓ | | COV-P (%) ↑ | | MAT-P (Å) ↓ | |
|---|---|---|---|---|---|---|---|---|---|---|
| | | | Mean | Median | Mean | Median | Mean | Median | Mean | Median |
| GeoDiff | 5000 | DDPM | 80.36 | 83.82 | 0.2820 | 0.2799 | 53.66 | 50.85 | 0.6673 | 0.4214 |
| SubgDiff | 5000 | DDPM (ours) | 90.91 | 95.59 | 0.2460 | 0.2351 | 50.16 | 48.01 | 0.6114 | 0.4791 |
| GeoDiff | 500 | DDPM | 80.20 | 83.59 | 0.3617 | 0.3412 | 45.49 | 45.45 | 1.1518 | 0.5087 |
| SubgDiff | 500 | DDPM (ours) | 89.78 | 94.17 | 0.2417 | 0.2449 | 50.03 | 48.31 | 0.5571 | 0.4921 |
| GeoDiff | 200 | DDPM | 69.90 | 72.04 | 0.4222 | 0.4272 | 36.71 | 33.51 | 0.8532 | 0.5554 |
| SubgDiff | 200 | DDPM (ours) | 85.53 | 88.99 | 0.2994 | 0.3033 | 47.76 | 45.89 | 0.6971 | 0.5118 |

**Domain generalization.** To further illustrate the benefits of SubgDiff, we design two cross-domain tasks: (1) Training on QM9 (small molecular with up to 9 heavy atoms) and testing on Drugs (medium-sized organic compounds); (2) Training on Drugs and testing on QM9. The results (Table 4 and Appendix Table 13) show that SubgDiff consistently outperforms GeoDiff and other models trained on the in-domain dataset, demonstrating the introduced sub-structure effectively enhances the robustness and generalization of the diffusion model.

Table 4: Results on the **GEOM-QM9** dataset for domain generalization. Except for GeoDiff and SubgDiff, the other methods are trained with in-domain data.

| Models | Train data | COV-R (%) ↑ Mean | COV-R (%) ↑ Median | MAT-R (Å) ↓ Mean | MAT-R (Å) ↓ Median |
|---|---|---|---|---|---|
| CVGAE [26] | QM9 | 0.09 | 0.00 | 1.6713 | 1.6088 |
| GraphDG [39] | QM9 | 73.33 | 84.21 | 0.4245 | 0.3973 |
| CGCF [50] | QM9 | 78.05 | 82.48 | 0.4219 | 0.3900 |
| ConfVAE [51] | QM9 | 77.84 | 88.20 | 0.4154 | 0.3739 |
| GeoMol [8] | QM9 | 71.26 | 72.00 | 0.3731 | 0.3731 |
| GeoDiff | Drugs | 74.94 | 79.15 | 0.3492 | 0.3392 |
| **SubgDiff** | Drugs | **83.50** | **88.70** | **0.3116** | **0.3075** |

## 6  Conclusion

We present a novel diffusion model SubgDiff, which involves the subgraph constraint in the diffusion model by introducing a mask vector to the forward process. Benefiting from the expectation state and $k$-step same-subgraph diffusion, SubgDiff effectively boosts the perception of molecular substructure in the denoising network, thereby achieving state-of-the-art performance at various downstream property prediction tasks. There are several exciting avenues for future work. The mask distribution can be made flexible such that more chemical prior knowledge may be incorporated into efficient subgraph sampling. Besides, the proposed SubgDiff can be generalized to proteins such that the denoising network can learn meaningful secondary structures.

## Acknowledgement

This project was supported by Shenzhen Hetao Shenzhen-Hong Kong Science and Technology Innovation Cooperation Zone, under Grant No. HTHZQSWS-KCCYB-2023052.

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

# Appendix

## Contents

## A  Experiment details and more results

### A.1  Visualization of representations

We conduct an alignment analysis to validate that our method can capture subgraphs during pretraining. Specifically, we employ t-distributed stochastic neighbor embedding (t-SNE) to visually represent molecules with various scaffolds. The purpose is to investigate whether molecules sharing the same scaffold exhibit similar representations, which are extracted by the pretrained molecular encoder. A scaffold, which is the core structure of a molecule, holds significant importance in the field of chemistry as it serves as a foundation for systematic exploration of molecular cores and building blocks. It is usually represented by a substructure of a molecule and can be regarded as the subgraph in our SubgDiff.

In our analysis, we select the nine most prevalent scaffolds from each dataset (BBBP, Sider, ClinTox, and Bace) and assign each molecule to a cluster according to its scaffold. To quantify the molecule embedding, we compute the Silhouette index of the embeddings for each dataset.

As shown in Table 5, SubgDiff enables the generation of more distinctive representations of molecules with different scaffolds. This implies that SubgDiff enables the denoising network (molecular encoder) to better capture the subgraph (scaffold) information. We also provide the t-SNE visualizations in Figure 6.

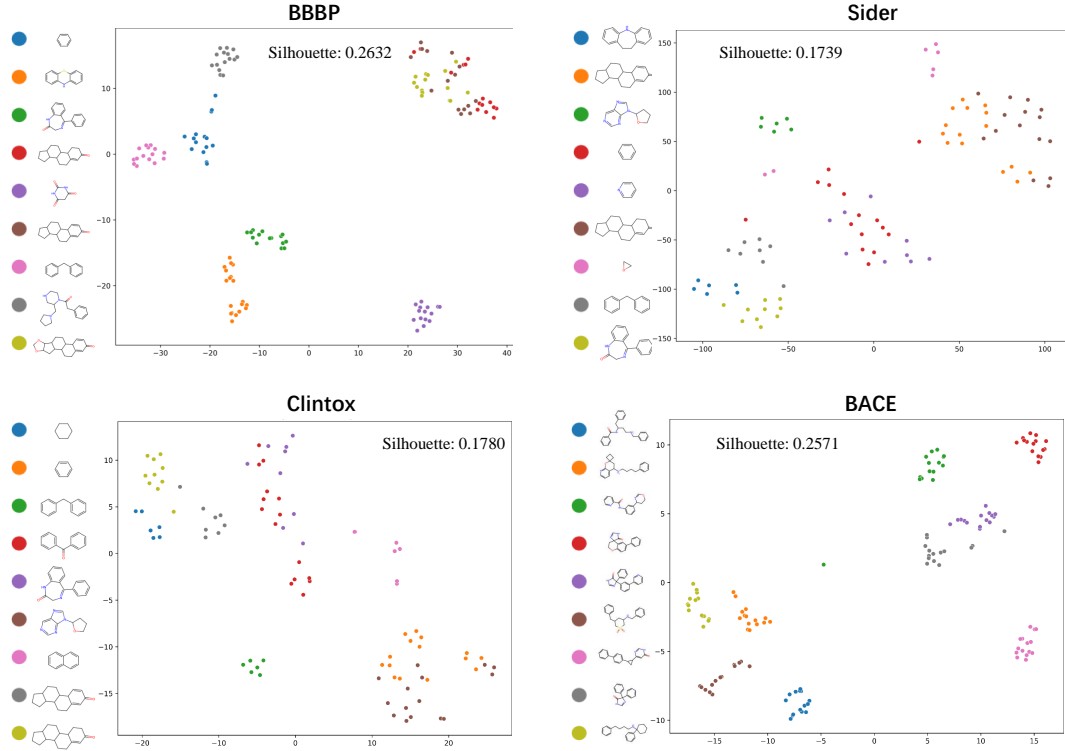

Figure 6: T-distributed stochastic neighbor embedding (t-SNE) visualization of the learned molecules representations, colored by the scaffolds of the molecules.

Table 5: Silhouette index (higher is better) of the molecule embeddings on Moleculenet dataset (with 2D topology only)

|  | BBBP ↑ | ToxCast ↑ | Sider ↑ | ClinTox ↑ | Bace ↑ |
|---|---|---|---|---|---|
| MoleculeSDE | 0.2344 | 0.0611 | 0.1664 | 0.1394 | 0.1860 |
| SubgDiff (ours) | **0.2632** | **0.0650** | **0.1739** | **0.1780** | **0.2571** |

## A.2 Mask distribution

In this paper, we pre-define the mask distribution to be a discrete distribution, with sample space $\chi = \{G_{sub}^i\}_{i=1}^N$, and $p_t(\mathcal{S} = G_{sub}^i) = 1/N, t > 1$, where $G_{sub}^i$ is the subgraph split by the Torsional-based decomposition methods [16]. The decomposition approach will cut off one torsional edge in a 3D molecule to make the molecule into two components, each of which contains at least two atoms. The two components are represented as two complementary mask vectors (i.e. $\mathbf{s}' + \mathbf{s} = \mathbf{1}$). Thus $n$ torsional edges in $G_{3D}^i$ will generate $2n$ subgraphs. Finally, for each atom $v$, the $s_{t_v} \sim Bern(0.5)$, i.e. $p = 0.5$ in SubgDiff.

## A.3 Hyperparameters

All models are trained with SGD using the ADAM optimizer. Pre-training takes around 24 hours with a single Nvidia A6000 GPU of 48GB RAM. The hyperparameters can be seen in Table 6 and Table 7.

Table 6: Additional hyperparameters of our SubgDiff.

| Task | $\beta_1$ | $\beta_T$ | $\beta$ scheduler | $T$ | k (k-same mask) | $\tau$ | Batch Size | Train Iter. |
|---|---|---|---|---|---|---|---|---|
| QM9 | 1e-7 | 2e-3 | sigmoid | 5000 | 250 | 10Å | 64 | 2M |
| Drugs | 1e-7 | 2e-3 | sigmoid | 5000 | 250 | 10Å | 32 | 6M |

Table 7: Additional hyperparameters of our SubgDiff with different timesteps.

| Task | $\beta_1$ | $\beta_T$ | $\beta$ scheduler | $T$ | k (k-same mask) | $\tau$ | Batch Size | Train Iter. |
|------|-----------|-----------|-------------------|-----|-----------------|--------|------------|-------------|
| 500-step QM9 | 1e-7 | 2e-2 | sigmoid | 500 | 25 | 10Å | 64 | 2M |
| 200-step QM9 | 1e-7 | 5e-2 | sigmoid | 200 | 10 | 10Å | 64 | 2M |
| 500-step Drugs | 1e-7 | 2e-2 | sigmoid | 500 | 25 | 10Å | 32 | 4M |
| 1000-step Drugs | 1e-7 | 9e-3 | sigmoid | 500 | 50 | 10Å | 32 | 4M |

## A.4 Settings and more results on molecular representation learning task

### A.4.1 Model architecture

We use the pretraining framework MoleculeSDE proposed by [23] and extend our SubgDiff to multi-modality pertaining. The two key components of MoleculeSDE are two SDEs(stochastic differential equations Song et al. [42]): an SDE from 2D topology to 3D conformation (2D → 3D) and an SDE from 3D conformation to 2D topology (3D → 2D). In practice, these two SDEs can be replaced by discrete diffusion models. In this paper, we use the proposed SubgDiff to replace the SDEs.

**2D topological molecular graph.** A topological molecular graph is denoted as $g_{2D} = \mathcal{G}(\mathcal{V}, \mathbb{E}, \mathbf{X})$, where $\mathbf{X}$ is the atom attribute matrix and $\mathbf{X}$ is the bond attribute matrix. The 2D graph representation with graph neural network (GNN) is:

$$\boldsymbol{x} \triangleq \boldsymbol{H}_{2D} = \text{GIN}(g_{2D}) = \text{GIN}(\mathcal{G}, \mathbf{X}), \tag{20}$$

where GIN is the a powerful 2D graph neural network [49] and $\boldsymbol{H}_{2D} = [h_{2D}^0, h_{2D}^1, \ldots]$, where $h_{2D}^i$ is the $i$-th node representation.

**3D conformational molecular graph.** The molecular conformation is denoted as $g_{3D} := G_{3D}(\mathcal{G}, \mathbf{R})$. The conformational representations are obtained by a 3D GNN SchNet [35]:

$$\boldsymbol{y} \triangleq \boldsymbol{H}_{3D} = \text{SchNet}(g_{3D}) = \text{SchNet}(\mathcal{G}, \mathbf{R}), \tag{21}$$

where $\boldsymbol{H}_{3D} = [h_{3D}^0, h_{3D}^1, \ldots]$, and $h_{3D}^i$ is the $i$-th node representation.

**An SE(3)-Equivariant Conformation Generation**   The first objective is the conditional generation from topology to conformation, $p(\boldsymbol{y}|\boldsymbol{x})$, implemented as SubgDiff. The denoising network we adopt is the SE(3)-equivariance network ($S_\theta^{2D\to3D}$) used in MoleculeSDE. The details of the network architecture refer to [23].

Therefore, the training objective from 2D topology graph to 3D confirmation is:

$$\mathcal{L}_{2D\to3D} = \mathbb{E}_{\boldsymbol{x}, \mathbf{R}, t, \mathbf{s}_t} \mathbb{E}_{\mathbf{R}_t|\mathbf{R}}$$
$$\left[ \left\| \text{diag}(\mathbf{s}_t)(\epsilon - S_\theta^{2D\to3D}(\boldsymbol{x}, \mathbf{R}_t, t)) \right\|_2^2 + \text{BCE}(\mathbf{s}_t, s_\vartheta^{2D\to3D}(\boldsymbol{x}, \mathbf{R}_t, t)) \right], \tag{22}$$

where $s_\vartheta^{2D\to3D}(\boldsymbol{x}, \mathbf{R}_t, t)$ gets the invariant feature from $S_\theta$ and introduces a mask head (MLP) to read out the mask prediction.

**An SE(3)-Invariant Topology Generation.**   The second objective is to reconstruct the 2D topology from 3D conformation, i.e., $p(\boldsymbol{x}|\boldsymbol{y})$. We also use the SE(3)-invariant score network $S_\theta^{3D\to2D}$ proposed by MoleculeSDE. The details of the network architecture refer to [23]. For modeling $S_\theta^{3D\to2D}$, it needs to satisfy the SE(3)-invariance symmetry property. The inputs are 3D conformational representation $\boldsymbol{y}$, the noised 2D information $\boldsymbol{x}_t$ at time $t$, and time $t$. The output of $S_\theta^{3D\to2D}$ is the Gaussian noise, as $(\epsilon^{\mathbf{X}}, \epsilon^{\mathbb{E}})$. The diffused 2D information contains two parts: $\boldsymbol{x}_t = (\mathbf{X}_t, \mathbb{E}_t)$. For node feature $\mathbf{X}$, the training objective is

$$\mathcal{L}_{3D\to2D}^{\mathbf{X}} = \mathbb{E}_{\mathbf{X}, \boldsymbol{y}} \mathbb{E}_{t, \mathbf{s}_t} \mathbb{E}_{\mathbf{X}_t|\mathbf{X}} \tag{23}$$
$$\left[ \left\| \text{diag}(\mathbf{s}_t)(\epsilon - S_\theta^{3D\to2D}(\boldsymbol{y}, \mathbf{X}_t, t)) \right\|_2^2 + \text{BCE}(\mathbf{s}_t, s_\vartheta^{3D\to2D}(\boldsymbol{y}, \mathbf{X}_t, t)) \right]. \tag{24}$$

For edge feature $\mathbb{E}$, we define a mask matrix $\mathbf{S}$ from mask vector $\mathbf{s}$: $\mathbf{S}_{ij} = 1$ if $\mathbf{s}_i = 1$ or $\mathbf{s}_j = 1$, otherwise, $\mathbf{S}_{ij} = 0$. Eventually, the ojective can be written as:

$$\mathcal{L}_{3D\to2D}^{\mathbb{E}} = \mathbb{E}_{\mathbb{E}, \boldsymbol{y}} \mathbb{E}_{t, \mathbf{s}_t} \mathbb{E}_{\mathbb{E}_t|\mathbb{E}} \tag{25}$$
$$\left[ \left\| \mathbf{S}_t \odot (\epsilon - S_\theta^{3D\to2D}(\boldsymbol{y}, \mathbb{E}_t, t)) \right\|_2^2 + \text{BCE}(\mathbf{s}_t, s_\vartheta^{3D\to2D}(\boldsymbol{y}, \mathbb{E}_t, t)) \right], \tag{26}$$

Then the score network $S_\theta^{3D \to 2D}$ is also decomposed into two parts for the atoms and bonds: $S_\theta^{\mathbf{X}_t}(\boldsymbol{x}_t)$ and $S_\theta^{\mathbb{E}_t}(\boldsymbol{x}_t)$. Similarly, the mask predictor $s_\vartheta^{3D \to 2D}(\boldsymbol{x}_t)$ is also decomposed into two parts for the atoms and bonds: $s_\vartheta^{\mathbf{X}_t}(\boldsymbol{x}_t)$ and $s_\vartheta^{\mathbb{E}_t}(\boldsymbol{x}_t)$.

Similar to the topology to conformation generation procedure, the $s_\vartheta^{3D \to 2D}(\boldsymbol{x}, \mathbf{R}_t, t)$ gets the invariant feature from $S_\theta^{3D \to 2D}$ and introduces a mask head (MLP) to read out the mask prediction.

**Learning.** Following MoleculeSDE, we incorporate a contrastive loss called EBM-NCE [22]. EBM-NCE provides an alternative approach to estimate the mutual information $I(X; Y)$ and is anticipated to complement the generative self-supervised learning (SSL) method. As a result, the ultimate objective is:

$$\mathcal{L}_{\text{overall}} = \alpha_1 \mathcal{L}_{\text{Contrastive}} + \alpha_2 \mathcal{L}_{2D \to 3D} + \alpha_3 (\mathcal{L}_{3D \to 2D}^{\mathbf{X}} + \mathcal{L}_{3D \to 2D}^{\mathbb{E}}), \tag{27}$$

where $\alpha_1, \alpha_2, \alpha_3$ are three coefficient hyperparameters.

### A.4.2  Dataset and settings

**Dataset.** For pretraining, following MoleculeSDE, we use PCQM4Mv2 [12]. It's a sub-dataset of PubChemQC [29] with 3.4 million molecules with both the topological graph and geometric conformations. For finetuning, in addition to QM9 [34], we also include MD17. To be specific, MD17 comprises eight molecular dynamics simulations focused on small organic molecules. These datasets were initially presented by Chmiela et al. [5] for the development of energy-conserving force fields using GDML. Each dataset features the trajectory of an individual molecule, encompassing a broad spectrum of conformations. The objective is to predict energies and forces for each trajectory by employing a single model.

**Baselines for 3D property prediction** We begin by incorporating three coordinate-MI-unaware SSL methods: (1) Type Prediction, which aims to predict the atom type of masked atoms; (2) Angle Prediction, which focuses on predicting the angle among triplet atoms, specifically the bond angle prediction; (3) 3D InfoGraph, which adopts the contrastive learning paradigm by considering the node-graph pair from the same molecule geometry as positive and negative otherwise. Next, in accordance with the work of [24], we include two contrastive baselines: (4) GeoSSL-InfoNCE [31] and (5) GeoSSL-EBM-NCE [22]. We also incorporate a generative SSL baseline named (6) GeoSSL-RR (RR for Representation Reconstruction). The above baselines are pre-trained on a subset of 1M molecules with 3D geometries from Molecule3D [53] and we reuse the results reported by [24] with SchNet as backbone.

**Baselines for 2D topology pretraining.** We pick up the most promising ones as follows. Attr-Mask [13, 21], ContexPred [13], InfoGraph [45], and MolCLR [47].

**Baselines for 2D and 3D multi-modality pretraining.** We include MoleculeSDE[23](Variance Exploding (VE) and Variance Preserving (VP)) as a crucial baseline to verify the effectiveness of our methods due to the same pertaining framework. We reproduce the results from the released Code.

### A.4.3  3D molecular property prediction Results on QM9.

By adopting the pertaining setting in Appendix A.4.2, we also take the QM9 dataset for finetuning and follow the literature [35, 36, 23], using 110K for training, 10K for validation and 11k for testing. In addition, the QM9 dataset encompasses 12 tasks that pertain to quantum properties, which are commonly used for evaluating representation learning tasks [35, 24]. The experimental results can be seen in Table 8. The results also suggest the superior performance of our method.

### A.4.4  Compared with GeoDiff.

We directly reuse the pre-trained model of the molecular conformation generation in Section 5.2 for fine-tuning, to compare our method with GeoDiff from naive denoising pretraining perspective [56]. The results are shown in Table 9.

Table 8: Results on 12 quantum mechanics prediction tasks from QM9. We take 110K for training, 10K for validation, and 11K for testing. The evaluation is mean absolute error (MAE), and the best and the second best results are marked in bold and underlined, respectively. The backbone is **SchNet**.

| Pretraining | Alpha↓ | Gap↓ | HOMO↓ | LUMO↓ | Mu↓ | Cv↓ | G298↓ | H298↓ | R2↓ | U298↓ | U0↓ | Zpve↓ |
|---|---|---|---|---|---|---|---|---|---|---|---|---|
| Random init | 0.070 | 50.59 | 32.53 | 26.33 | 0.029 | 0.032 | 14.68 | 14.85 | 0.122 | 14.70 | 14.44 | 1.698 |
| Supervised | 0.070 | 51.34 | 32.62 | 27.61 | 0.030 | 0.032 | 14.08 | 14.09 | 0.141 | 14.13 | 13.25 | 1.727 |
| Type Prediction | 0.084 | 56.07 | 34.55 | 30.65 | 0.040 | 0.034 | 18.79 | 19.39 | 0.201 | 19.29 | 18.86 | 2.001 |
| Angle Prediction | 0.084 | 57.01 | 37.51 | 30.92 | 0.037 | 0.034 | 15.81 | 15.89 | 0.149 | 16.41 | 15.76 | 1.850 |
| 3D InfoGraph | 0.076 | 53.33 | 33.92 | 28.55 | 0.030 | 0.032 | 15.97 | 16.28 | 0.117 | 16.17 | 15.96 | 1.666 |
| GeossL-RR | 0.073 | 52.57 | 34.44 | 28.41 | 0.033 | 0.038 | 15.74 | 16.11 | 0.194 | 15.58 | 14.76 | 1.804 |
| GeossL-InfoNCE | 0.075 | 53.00 | 34.29 | 27.03 | 0.029 | 0.033 | 15.67 | 15.53 | 0.125 | 15.79 | 14.94 | 1.675 |
| GeossL-EBM-NCE | 0.073 | 52.86 | 33.74 | 28.07 | 0.031 | 0.032 | 14.02 | 13.65 | 0.121 | 13.70 | 13.45 | 1.677 |
| MoleculeSDE | 0.062 | 47.74 | 28.02 | 24.60 | 0.028 | 0.029 | 13.25 | 12.70 | 0.120 | 12.68 | 12.93 | 1.643 |
| **Ours** | **0.054** | **44.88** | **25.45** | **23.75** | **0.027** | **0.028** | **12.03** | **11.46** | **0.110** | **11.32** | **11.25** | **1.568** |

Table 9: Results on 12 quantum mechanics prediction tasks from QM9. We take 110K for training, 10K for validation, and 11K for testing. The evaluation is mean absolute error (MAE), and the best and the second best results are marked in bold and underlined, respectively. The backbone is **SchNet**.

| Pretraining | Alpha↓ | Gap↓ | HOMO↓ | LUMO↓ | Mu↓ | Cv↓ | G298↓ | H298↓ | R2↓ | U298↓ | U0↓ | Zpve↓ |
|---|---|---|---|---|---|---|---|---|---|---|---|---|
| GeoDiff | 0.078 | 51.84 | 30.88 | 28.29 | 0.028 | 0.035 | 15.35 | 11.37 | 0.132 | 15.76 | 15.24 | 1.869 |
| **SubgDiff (ours)** | 0.076▲ | 50.80▲ | 31.15▼ | 26.62▲ | 0.025▲ | 0.032▲ | 14.92▲ | 12.86▲ | 0.129▲ | 14.74▲ | 14.53▲ | 1.710▲ |

## A.5 Settings and more results on conformation generation task

### A.5.1 Dataset and network.

Following prior works [52], we utilize the GEOM-QM9 [34] and GEOM-Drugs [2] datasets. The former dataset comprises small molecules of up to 9 heavy atoms, while the latter contains larger drug-like compounds. We reuse the data split provided by Xu et al. [52]. For both datasets, the training dataset comprises $40,000$ molecules, each with 5 conformations, resulting in $200,000$ conformations in total. The test split includes 200 distinctive molecules, with $14,324$ conformations for Drugs and $22,408$ conformations for QM9.

We adopt the graph field network (GFN) from [52] as the GNN encoder for extracting the 3D molecular information. In the $l$-th layer, the GFN receives node embeddings $\mathbf{h}^l \in \mathbb{R}^{n \times b}$ (where $b$ represents the feature dimension) and corresponding coordinate embeddings $\mathbf{x}^l \in \mathbb{R}^{n \times 3}$ as input. It then produces the output $\mathbf{h}^{l+1}$ and $\mathbf{x}^{l+1}$ according to the following process:

$$\mathbf{m}_{ij}^l = \Phi_m^l \left( \mathbf{h}_i^l, \mathbf{h}_j^l, \|\mathbf{x}_i^l - \mathbf{x}_j^l\|^2, e_{ij}; \theta_m \right) \tag{28}$$

$$\mathbf{h}_i^{l+1} = \Phi_h^l \left( \mathbf{h}_i^l, \sum_{j \in \mathcal{N}(i)} \mathbf{m}_{ij}^l; \theta_h \right) \tag{29}$$

$$\mathbf{x}_i^{l+1} = \sum_{j \in \mathcal{N}(i)} \frac{1}{d_{ij}} \left( R_i - R_j \right) \Phi_x^l \left( \mathbf{m}_{ij}^l; \theta_x \right) \tag{30}$$

where $\Phi$ are implemented as feed-forward networks and $d_{ij}$ denotes interatomic distances. The initial embedding $\mathbf{h}^0$ is composed of atom embedding and time step embedding while $\mathbf{x}^0$ represents atomic coordinates. $\mathcal{N}(i)$ is the neighborhood of $i^{th}$ node, consisting of connected atoms and other ones within a radius threshold $\tau$, helping the model capture long-range interactions explicitly and support disconnected molecular graphs.

Eventually, the Gaussian noise and mask can be predicted as follows (C.f. Figure 7):

$$\hat{\epsilon}_i = \mathbf{x}_i^L \tag{31}$$

$$\hat{s}_i = \text{MLP}(\mathbf{h}_i^L) \tag{32}$$

where $\hat{\epsilon}_i$ is equivalent and $\hat{s}_i$ is invariant.

**Settings**. For GeoDiff [52] with 5000 steps, we use the checkpoints released in public GitHub to reproduce the results. For 200 and 500 steps, we retrain it and do the DDPM sampling.

### A.5.2 Evaluation metrics for conformation generation.

To compare the generated and ground truth conformer ensembles, we employ the same evaluation metrics as in a prior study [8]: Average Minimum RMSD (AMR) and Coverage. These metrics

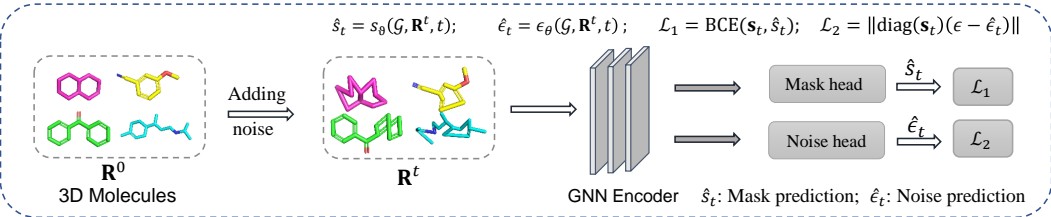

Figure 7: The model architecture for denoising SubgDiff.

enable us to assess the quality of the generated conformers from two perspectives: Recall (R) and Precision (P). Recall measures the extent to which the generated ensemble covers the ground-truth ensemble, while Precision evaluates the accuracy of the generated conformers.

The four metrics built upon root-mean-square deviation (RMSD), which is defined as the normalized Frobenius norm of two atomic coordinates matrices, after alignment by Kabsch algorithm [17]. Formally, let $S_g$ and $S_r$ denote the sets of generated and reference conformers respectively, then the **Cov**erage and **Mat**ching metrics [50] can be defined as:

$$\text{COV-R}(S_g, S_r) = \frac{1}{|S_r|} \left| \left\{ \mathcal{C} \in S_r | \text{RMSD}(\mathcal{C}, \hat{\mathcal{C}}) \leq \delta, \hat{\mathcal{C}} \in S_g \right\} \right|, \tag{33}$$

$$\text{MAT-R}(S_g, S_r) = \frac{1}{|S_r|} \sum_{\mathcal{C} \in S_r} \min_{\hat{\mathcal{C}} \in S_g} \text{RMSD}(\mathcal{C}, \hat{\mathcal{C}}), \tag{34}$$

where $\delta$ is a threshold. The other two metrics COV-P and MAT-P can be defined similarly but with the generated sets $S_g$ and reference sets $S_r$ exchanged.

$$\text{COV-P}(S_r, S_g) = \frac{1}{|S_g|} \left| \left\{ \hat{\mathcal{C}} \in S_g | \text{RMSD}(\mathcal{C}, \hat{\mathcal{C}}) \leq \delta, \mathcal{C} \in S_r \right\} \right|, \tag{35}$$

$$\text{MAT-P}(S_r, S_g) = \frac{1}{|S_g|} \sum_{\hat{\mathcal{C}} \in S_g} \min_{\mathcal{C} \in S_r} \text{RMSD}(\mathcal{C}, \hat{\mathcal{C}}), \tag{36}$$

In practice, $S_g$ is set as twice of the size of $S_r$ for each molecule.

### A.5.3 Comparison with GeoDiff using Langevin Dynamics sampling method.

In order to verify that our proposed diffusion process can bring benefits to other sampling methods, we conduct the experiments to compare our proposed diffusion model with GeoDiff by adopting a typical sampling method Langevin dynamics (LD sampling)[41] :

$$\mathbf{R}^{t-1} = \mathbf{R}^t + \alpha_t \epsilon_\theta(\mathcal{G}, \mathbf{R}^t, t) + \sqrt{2\alpha_t} \mathbf{z}_{t-1} \tag{37}$$

where $\mathbf{z}_t \sim \mathcal{N}(\mathbf{0}, \mathbf{I})$ and $h\sigma_t^2$. $h$ is the hyper-parameter referring to step size and $\sigma_t$ is the noise schedule in the forward process. We use various time-step to evaluate the generalization and robustness of the proposed method, and the results shown in Table 10 indicate that our method significantly outperforms GeoDiff, especially when the time-step is relatively small (200,500), which implies that our training method can effectively improve the efficiency of denoising. Similar results are also observed on the GEOM-drugs dataset (Table 11).

Table 10: Results on **GEOM-QM9** dataset with different time steps. Langevin dynamics [41] is a typical sampling method used in DPM. ▲denotes SubgDiff outperforms GeoDiff. The threshold $\delta = 0.5$Å.

| Steps | Sampling method | Models | COV-R (%) ↑ | | MAT-R (Å) ↓ | | COV-P (%) ↑ | | MAT-P (Å) ↓ | |
|---|---|---|---|---|---|---|---|---|---|---|
| | | | Mean | Median | Mean | Median | Mean | Median | Mean | Median |
| 500 | Langevin dynamics | GeoDiff | 87.80 | 93.66 | 0.3179 | 0.3216 | 46.25 | 45.02 | 0.6173 | 0.5112 |
| 500 | Langevin dynamics | SubgDiff | 91.40▲ | 95.39▲ | 0.2543▲ | 0.2601▲ | 51.71▲ | 48.50▲ | 0.5035▲ | 0.4734▲ |
| 200 | Langevin dynamics | GeoDiff | 86.60 | 93.09 | 0.3532 | 0.3574 | 42.98 | 42.60 | 0.5563 | 0.5367 |
| 200 | Langevin dynamics | SubgDiff | 90.36▲ | 95.93▲ | 0.3064▲ | 0.3098▲ | 48.56▲ | 46.46▲ | 0.5540▲ | 0.5082▲ |

Table 11: Results on the **GEOM-Drugs** dataset under different diffusion timesteps. DDPM [10] is the sampling method used in GeoDiff and Langevin dynamics [41] is a typical sampling method used in DPM. Our proposed sampling method (Algorithm 2) can be viewed as a DDPM variant. ▲/▼ denotes SubgDiff outperforms/underperforms GeoDiff. The threshold $\delta = 1.25$Å.

| Models | Timesteps | Sampling method | COV-R (%) ↑ | | MAT-R (Å) ↓ | |
| | | | Mean | Median | Mean | Median |
| --- | --- | --- | --- | --- | --- | --- |
| GeoDiff | 500 | DDPM | 50.25 | 48.18 | 1.3101 | 1.2967 |
| SubgDiff | 500 | DDPM (ours) | 76.16▲ | 86.43▲ | 1.0463▲ | 1.0264▲ |
| GeoDiff | 500 | LD | 64.12 | 75.56 | 1.1444 | 1.1246 |
| SubgDiff | 500 | LD (ours) | 74.30▲ | 77.87▲ | 1.0003▲ | 0.9905▲ |

Table 12: Results on **GEOM-QM9** dataset. The threshold $\delta = 0.5$Å.

| Models | COV-R (%) ↑ | | MAT-R (Å) ↓ | | COV-P (%) ↑ | | MAT-P (Å) ↓ | |
| | Mean | Median | Mean | Median | Mean | Median | Mean | Median |
| --- | --- | --- | --- | --- | --- | --- | --- | --- |
| CVGAE | 0.09 | 0.00 | 1.6713 | 1.6088 | - | - | - | - |
| GraphDG | 73.33 | 84.21 | 0.4245 | 0.3973 | 43.90 | 35.33 | 0.5809 | 0.5823 |
| CGCF | 78.05 | 82.48 | 0.4219 | 0.3900 | 36.49 | 33.57 | 0.6615 | 0.6427 |
| ConfVAE | 77.84 | 88.20 | 0.4154 | 0.3739 | 38.02 | 34.67 | 0.6215 | 0.6091 |
| GeoMol | 71.26 | 72.00 | 0.3731 | 0.3731 | - | - | - | - |
| ConfGF | 88.49 | 94.31 | 0.2673 | 0.2685 | 46.43 | 43.41 | **0.5224** | 0.5124 |
| GeoDiff | 80.36 | 83.82 | 0.2820 | 0.2799 | **53.66** | **50.85** | 0.6673 | **0.4214** |
| **SubgDiff** | **90.91** | **95.59** | **0.2460** | **0.2351** | 50.16 | 48.01 | 0.6114 | 0.4791 |

## A.5.4 Comparison with SOTAs.

**i) Baselines:** We compare SubgDiff with 7 state-of-the-art baselines: CVGAE [26], GraphDG [39], CGCF [50], ConfVAE [51], ConfGF [38] and GeoDiff [52]. For the above baselines, we reuse the experimental results reported by [52]. For GeoDiff [52], we use the checkpoints released in public GitHub to reproduce the results. **ii) Results:** The results on the GEOM-QM9 dataset are reported in Table 12. From the results, we get the following observation: SubgDiff significantly outperforms the baselines on COV-R, indicating the SubgDiff tends to explore more possible conformations. This implicitly demonstrates the subgraph will help fine-tune the generated conformation to be a potential conformation.

## A.5.5 Domain generelizaion

The results of Training on QM9 (small molecular with up to 9 heavy atoms) and testing on Drugs (medium-sized organic compounds) can be found in table 13.

Table 13: Results on the **GEOM-Drugs** dataset. The threshold $\delta = 1.25$Å

| Models | Train data | COV-R (%) ↑ | | MAT-R (Å) ↓ | |
| | | Mean | Median | Mean | Median |
| --- | --- | --- | --- | --- | --- |
| CVGAE | Drugs | 0.00 | 0.00 | 3.0702 | 2.9937 |
| GraphDG | Drugs | 8.27 | 0.00 | 1.9722 | 1.9845 |
| GeoDiff | QM9 | 7.99 | 0.00 | 2.7704 | 2.3297 |
| **SubgDiff** | QM9 | **24.01** | **9.93** | **1.6128** | **1.5819** |

## A.6 Sentivity analysis of $k$ in $k$-step same-subgraph diffusion

The results from Table 14 indicate that $k = 25$ is more likely to give the best performance when the diffusion step $N$ is 500. From our experience, $k$ and the number of the diffusion step $N$ can maintain a certain ratio, e.g. $N : k = 20$ in our experiments, and the model can achieve better performance.

Table 14: The sensitivity analysis for different $k$ in $k$-step same subgraph diffusion on conformation generation

| $k$ | 1ld sampling* | 10 | 25 | 50 |
|---|---|---|---|---|
| COV-R(Mean) ↑ | 89.70 | 88.06 | **89.78** | 89.02 |
| COV-R(Median) ↑ | 93.96 | 93.26 | **94.17** | 93.21 |
| COV-P(Mean) (%) ↑ | 49.90 | 47.38 | **50.03** | 48.63 |
| COV-P(Median) (%) ↑ | 47.00 | 47.06 | **48.31** | 46.77 |
| MAT-R(Mean) (A) ↓ | 0.5235 | 0.2623 | **0.2417** | 0.2706 |
| MAT-R(Median) (A) ↓ | 0.2710 | 0.2597 | **0.2449** | 0.2709 |
| MAT-P(Mean) (A) ↓ | 4.7816 | 4.2922 | **0.5571** | 0.7512 |
| MAT-P(Median) (A) ↓ | 0.5378 | 0.5615 | **0.4921** | 0.4995 |

*The subgraph predictor cannot predict the correct subgraphs so we use Langevin Dynamics sampling rather than DDPM when $k = 1$

# B  More discussions

## B.1  More related works

**Masks on diffusion models.**     Previous works also share a similar idea of subgraph (mask) diffusion, such as MDM [32], MDSM [19] and SSSD [1]. However, the difference between our SubgDiff and theirs mainly lies in the following two aspects: i) Usage: the mask matrix/vector in SSSD and MDSM is fixed in all training steps, which means some segments of the data (time series or images) will never be diffused. But our method samples the $\mathbf{s}_t \sim p_{\mathbf{s}_t}(\mathcal{S})$ at each time step, hence a suitable discrete distribution $p(\mathcal{S})$ can ensure that almost all nodes can be added noise. ii) Purpose: MDSM and MDM concentrate on self-supervised pre-training, while SubgDiff serves as a potent generative model and self-supervised pre-training algorithm. Notably, when $\mathbf{s}_t = \mathbf{s}_0, \forall t$, SubgDiff can recover to MDSM.

**Graph generation models.**     D3FG [20]: D3FG adopts three different diffusion models (D3PM, DDPM, and SO(3) Diffusion) to generate three different parts of molecules(linkerr types, center atom position, and functional group orientations), respectively. In general, these three parts can also be viewed as three subgraphs(subset). DiffPACK[60] is an Autoregressive generative method that predicts the torsional angle $\chi_i (i = 1, 2, .., 4)$ of protein side-chains with the condition $\chi_{1,...,i-1}$, where $\chi_i$ is a predefined subset of atoms. It uses a torsional-based diffusion model to approximate the distribution $p(\chi_i|\chi_{1,...,i-1})$, in which every subset $\chi_i$ needs a separate score network to estimate. Essentially, both D3FG and DiffPACK can be viewed as selecting a subset first and then only adding noise on the fixed subset during the entire diffusion process. In contrast, our method proposes to randomly sample a subset from mask distribution $p(S)$ in *each time-step* during the forward process. [18] proposes an autoregressive diffusion model named GraphARM, which absorbs one node in each time step by masking it along with its connecting edges during the forward process. Differently from GraphARM, our SubgDiff selects a subgraph in each time step to inject the Gaussian noise, which is equivalent to masking several nodes during the forward process. In addition, the number of steps in GraphARM must be the same as the number of nodes due to the usage of the absorbing state, while our method can set any time-step during diffusion theoretically since we use the real-value Gaussian noise. Concurrently, SubDiff [54] is proposed to use subgraphs as minimum units to train a latent diffusion model, while our method directly involves the subgraph during the forward process, which is a new type of diffusion model.

## B.2  Limitations

One limitation of our method is that it does not currently align with the Boltzmann distribution over the conformations and it cannot accurately estimate the likelihood of a sampled conformation. Currently, we generate the subgraph by dividing the molecule with the rotatable bonds. More chemical prior knowledge can be used to guide the mask of the subgraph. Additionally, there is a requirement for the number of steps in sampling to be the same as that in training, which can be improved with the recent progress of diffusion models such as DDIM.

## C  An important lemma for diffusion model

According to [40, 10], the diffusion model is trained by optimizing the variational bound on the negative log-likelihood $-\log p_\theta(R^0)$, in which the tricky terms are $L_{t-1} = D_{KL}(q(R^{t-1}|R^t, R^0) \| p_\theta(R^{t-1}|R^t)))$, $T \geq t > 1$. Here we provide a lemma that tells us the posterior distribution $q(R^{t-1}|R^t, R^0)$ used in the training and sampling algorithms of the diffusion model can be determined by $q(R^t|R^{t-1}, R^0)$, $q(R^{t-1}|R^0)$. Formally, we have

**Lemma C.1** *Assume the forward and reverse processes of the diffusion model are both Markov chains. Given the forward Gaussian distribution* $q(R^t|R^{t-1}, R^0) = \mathcal{N}(R^t; \mu_1 R^{t-1}, \sigma_1^2 \boldsymbol{I})$, $q(R^{t-1}|R^0) = \mathcal{N}(R^{t-1}; \mu_2 R^0, \sigma_2^2 \boldsymbol{I})$ *and* $\epsilon_0 \sim \mathcal{N}(\mathbf{0}, \mathbf{I})$, *the distribution* $q(R^{t-1}|R^t, R^0)$ *is*

$$q(R^{t-1}|R^t, R^0) \propto \mathcal{N}(R^{t-1}; \frac{1}{\mu_1}(R^t - \frac{\sigma_1^2}{\sqrt{\mu_1^2\sigma_2^2 + \sigma_1^2}}\epsilon_0), \frac{\sigma_1^2\sigma_2^2}{\mu_1^2\sigma_2^2 + \sigma_1^2}\boldsymbol{I}).$$

*Parameterizing* $p_\theta(R^{t-1}|R^t)$ *in the reverse process as*

$$\mathcal{N}(R^{t-1}; \frac{1}{\mu_1}(R^t - \frac{\sigma_1^2}{\sqrt{\mu_1^2\sigma_2^2 + \sigma_1^2}}\epsilon_\theta(R^t, t)), \frac{\sigma_1^2\sigma_2^2}{\mu_1^2\sigma_2^2 + \sigma_1^2}\boldsymbol{I}),$$

*the training objective of the DPM can be written as*

$$\mathcal{L}(\theta) = \mathbb{E}_{t,R^0,\epsilon}\left[\frac{\sigma_1^2}{2\mu_1^2\sigma_2^2}\|\epsilon - \epsilon_\theta(\mu_1\mu_2 R^0 + \sqrt{\mu_1^2\sigma_2^2 + \sigma_1^2}\epsilon, t)\|^2\right],$$

*and the sampling process is*

$$R^{t-1} = \frac{1}{\mu_1}\left(R^t - \frac{\sigma_1^2}{\sqrt{\mu_1^2\sigma_2^2 + \sigma_1^2}}\epsilon_\theta(R^t, t)\right) + \frac{\sigma_1\sigma_2}{\sqrt{\mu_1^2\sigma_2^2 + \sigma_1^2}}z, \tag{38}$$

*where* $z \sim \mathcal{N}(\mathbf{0}, \mathbf{I})$.

Once we get the variables $(\mu_1, \sigma_1, \mu_2, \sigma_2)$, we can directly obtain the training objective and sampling process via lemma C.1, which will help the design of new diffusion models.

**Proof:**  Given the forward Gaussian distribution $q(R^t|R^{t-1}, R^0) = \mathcal{N}(R^t; \mu_1 R^{t-1}, \sigma_1^2 I)$ and $q(R^{t-1}|R^0) = \mathcal{N}(R^{t-1}; \mu_2 R^0, \sigma_2^2 I)$, we have

$$q(R^t|R^0) = q(R^t|R^{t-1}, R^0)q(R^{t-1}|R^0) = \mathcal{N}(R^t; \mu_1\mu_2 R^0, (\sigma_1^2 + \mu_1^2\sigma_2^2)I) \tag{39}$$

From the DDPM, we know training a diffusion model should optimize the ELBO of the data

$$\log p(\mathbf{R}) \geq \mathbb{E}_{q(\mathbf{R}^{1:T}|\mathbf{R}^0)}\left[\log \frac{p(\mathbf{R}^{0:T})}{q(\mathbf{R}^{1:T}|\mathbf{R}^0)}\right] \tag{40}$$

$$= \underbrace{\mathbb{E}_{q(\mathbf{R}^1|\mathbf{R}^0)}\left[\log p_\theta(\mathbf{R}^0|\mathbf{R}^1)\right]}_{\text{reconstruction term}} - \underbrace{D_{\text{KL}}(q(\mathbf{R}^T|\mathbf{R}^0) \| p(\mathbf{R}^T))}_{\text{prior matching term}} - \sum_{t=2}^{T}\underbrace{\mathbb{E}_{q(\mathbf{R}^t|\mathbf{R}^0)}\left[D_{\text{KL}}(q(\mathbf{R}^{t-1}|\mathbf{R}^t, \mathbf{R}^0) \| p_\theta(\mathbf{R}^{t-1}|\mathbf{R}^t))\right]}_{\text{denoising matching term}} \tag{41}$$

To compute the KL divergence $D_{\text{KL}}(q(\mathbf{R}^{t-1}|\mathbf{R}^t, \mathbf{R}^0) \parallel p_{\boldsymbol{\theta}}(\mathbf{R}^{t-1}|\mathbf{R}^t))$, we first rewrite $q(\mathbf{R}^{t-1}|\mathbf{R}^t, \mathbf{R}^0)$ by Bayes rule

$$q(R^{t-1}|R^t, R^0) = \frac{q(R^t|R^{t-1}, R^0)q(R^{t-1}|R^0)}{q(R^t|R^0)} \tag{42}$$

$$= \frac{\mathcal{N}(R^t; \mu_1 R^{t-1}, \sigma_1^2 \mathbf{I})\mathcal{N}(R^{t-1}; \mu_2 R^0, \sigma_2^2 \mathbf{I})}{\mathcal{N}(R^t; \mu_1 \mu_2 R^0, (\sigma_1^2 + \mu_1^2 \sigma_2^2)\mathbf{I})} \tag{43}$$

$$\propto \exp\left\{-\left[\frac{(R^t - \mu_1 R^{t-1})^2}{2\sigma_1^2} + \frac{(R^{t-1} - \mu_2 R^0)^2}{2\sigma_2^2} - \frac{(R^t - \mu_1 \mu_2 R^0)^2}{2(\sigma_1^2 + \mu_1^2 \sigma_2^2)}\right]\right\} \tag{44}$$

$$= \exp\left\{-\frac{1}{2}\left[\frac{(R^t - \mu_1 R^{t-1})^2}{\sigma_1^2} + \frac{(R^{t-1} - \mu_2 R^0)^2}{\sigma_2^2} - \frac{(R^t - \mu_1 \mu_2 R^0)^2}{\sigma_1^2 + \mu_1^2 \sigma_2^2}\right]\right\} \tag{45}$$

$$= \exp\left\{-\frac{1}{2}\left[\frac{(-2\mu_1 R^t R^{t-1} + \mu_1^2 (R^{t-1})^2)}{\sigma_1^2} + \frac{((R^{t-1})^2 - 2\mu_2 R^{t-1} R^0)}{\sigma_2^2} + C(R^t, R^0)\right]\right\} \tag{46}$$

$$\propto \exp\left\{-\frac{1}{2}\left[-\frac{2\mu_1 R^t R^{t-1}}{\sigma_1^2} + \frac{\mu_1^2 (R^{t-1})^2}{\sigma_1^2} + \frac{(R^{t-1})^2}{\sigma_2^2} - \frac{2\mu_2 R^{t-1} R^0}{\sigma_2^2}\right]\right\} \tag{47}$$

$$= \exp\left\{-\frac{1}{2}\left[(\frac{\mu_1^2}{\sigma_1^2} + \frac{1}{\sigma_2^2})(R^{t-1})^2 - 2\left(\frac{\mu_1 R^t}{\sigma_1^2} + \frac{\mu_2 R^0}{\sigma_2^2}\right) R^{t-1}\right]\right\} \tag{48}$$

$$= \exp\left\{-\frac{1}{2}\left[\frac{\sigma_1^2 + \mu_1^2 \sigma_2^2}{\sigma_1^2 \sigma_2^2}(R^{t-1})^2 - 2\left(\frac{\mu_1 R^t}{\sigma_1^2} + \frac{\mu_2 R^0}{\sigma_2^2}\right) R^{t-1}\right]\right\} \tag{49}$$

$$= \exp\left\{-\frac{1}{2}\left(\frac{\sigma_1^2 + \mu_1^2 \sigma_2^2}{\sigma_1^2 \sigma_2^2}\right)\left[(R^{t-1})^2 - 2\frac{\left(\frac{\mu_1 R^t}{\sigma_1^2} + \frac{\mu_2 R^0}{\sigma_2^2}\right)}{\frac{\sigma_1^2 + \mu_1^2 \sigma_2^2}{\sigma_1^2 \sigma_2^2}} R^{t-1}\right]\right\} \tag{50}$$

$$= \exp\left\{-\frac{1}{2}\left(\frac{\sigma_1^2 + \mu_1^2 \sigma_2^2}{\sigma_1^2 \sigma_2^2}\right)\left[(R^{t-1})^2 - 2\frac{\left(\frac{\mu_1 R^t}{\sigma_1^2} + \frac{\mu_2 R^0}{\sigma_2^2}\right)\sigma_1^2 \sigma_2^2}{\sigma_1^2 + \mu_1^2 \sigma_2^2} R^{t-1}\right]\right\} \tag{51}$$

$$= \exp\left\{-\frac{1}{2}\left(\frac{1}{\frac{\sigma_1^2 \sigma_2^2}{\sigma_1^2 + \mu_1^2 \sigma_2^2}}\right)\left[(R^{t-1})^2 - 2\frac{\mu_1 \sigma_2^2 R^t + \mu_2 \sigma_1^2 R^0}{\sigma_1^2 + \mu_1^2 \sigma_2^2} R^{t-1}\right]\right\} \tag{52}$$

$$\propto \mathcal{N}(R^{t-1}; \underbrace{\frac{\mu_1 \sigma_2^2 R^t + \mu_2 \sigma_1^2 R^0}{\sigma_1^2 + \mu_1^2 \sigma_2^2}}_{\mu_q(R^t, R^0)}, \underbrace{\frac{\sigma_1^2 \sigma_2^2}{\sigma_1^2 + \mu_1^2 \sigma_2^2}}_{\boldsymbol{\Sigma}_q(t)}\mathbf{I}) \tag{53}$$

We can rewrite our variance equation as $\boldsymbol{\Sigma}_q(t) = \sigma_q^2(t)\mathbf{I}$, where:

$$\sigma_q^2(t) = \frac{\sigma_1^2 \sigma_2^2}{\sigma_1^2 + \mu_1^2 \sigma_2^2} \tag{54}$$

From (39), we have the relationship between $R^t$ and $R^0$:

$$R^0 = \frac{R^t - \sqrt{\sigma_1^2 + \mu_1^2 \sigma_2^2}\epsilon}{\mu_1 \mu_2} \tag{55}$$

Substituting this into $\mu_q(R^t, R^0)$, we can get

$$\mu_q(R^t, R^0) = \frac{\mu_1 \sigma_2^2 R^t + \mu_2 \sigma_1^2 R^0}{\sigma_1^2 + \mu_1^2 \sigma_2^2} \tag{56}$$

$$= \frac{\mu_1 \sigma_2^2 R^t + \mu_2 \sigma_1^2 \frac{R^t - \sqrt{\sigma_1^2 + \mu_1^2 \sigma_2^2}\epsilon}{\mu_1 \mu_2}}{\sigma_1^2 + \mu_1^2 \sigma_2^2} \tag{57}$$

$$= \frac{\mu_1 \sigma_2^2 R^t + \frac{\sigma_1^2 R^2}{\mu_1} - \frac{\sigma_1^2 \sqrt{\sigma_1^2 + \mu_1^2 \sigma_2^2}\epsilon}{\mu_1}}{\sigma_1^2 + \mu_1^2 \sigma_2^2} \tag{58}$$

$$= \frac{1}{\mu_1} R^t - \frac{\sigma_1^2}{\mu_1 \sqrt{\sigma_1^2 + \mu_1^2 \sigma_2^2}}\epsilon \tag{59}$$

Thus,

$$q(R^{t-1}|R^t, R^0) \propto \mathcal{N}(R^{t-1}; \underbrace{\frac{1}{\mu_1}(R^t - \frac{\sigma_1^2}{\sqrt{\sigma_1^2 + \mu_1^2\sigma_2^2}}\epsilon)}_{\mu_q(R^t,t)}, \underbrace{\frac{\sigma_1^2\sigma_2^2}{\sigma_1^2 + \mu_1^2\sigma_2^2}\mathbf{I}}_{\mathbf{\Sigma}_q(t)}) \tag{60}$$

Parameterizing $p_\theta(R^{t-1}|R^t)$ in the reverse process as $\mathcal{N}(R^{t-1}; \frac{1}{\mu_1}(R^t - \frac{\sigma_1^2}{\sqrt{\mu_1^2\sigma_2^2 + \sigma_1^2}}\epsilon_\theta(R^t, t)), \frac{\sigma_1^2\sigma_2^2}{\mu_1^2\sigma_2^2 + \sigma_1^2}\mathbf{I})$ , and the corresponding optimization problem becomes:

$$\arg\min_\theta D_{\text{KL}}(q(R^{t-1}|R^t, R^0) \| p_\theta(R^{t-1}|R^t))$$

$$= \arg\min_\theta D_{\text{KL}}(\mathcal{N}\left(R^{t-1}; \mu_q, \Sigma_q(t)\right) \| \mathcal{N}\left(R^{t-1}; \mu_\theta, \Sigma_q(t)\right)) \tag{61}$$

$$= \arg\min_\theta \frac{1}{2\sigma_q^2(t)} \left[ \left\| \frac{\sigma_1^2}{\mu_1\sqrt{\sigma_1^2 + \mu_1^2\sigma_2^2}}\epsilon_0 - \frac{\sigma_1^2}{\mu_1\sqrt{\sigma_1^2 + \mu_1^2\sigma_2^2}}\epsilon_\theta(R^t, t) \right\|_2^2 \right] \tag{62}$$

$$= \arg\min_\theta \frac{1}{2\sigma_q^2(t)} \left[ \left\| \frac{\sigma_1^2}{\mu_1\sqrt{\sigma_1^2 + \mu_1^2\sigma_2^2}}(\epsilon_0 - \hat{\epsilon}_\theta(R^t, t)) \right\|_2^2 \right] \tag{63}$$

$$= \arg\min_\theta \frac{1}{2\sigma_q^2(t)} \left( \frac{\sigma_1^2}{\mu_1\sqrt{\sigma_1^2 + \mu_1^2\sigma_2^2}} \right)^2 \left[ \|\epsilon_0 - \hat{\epsilon}_\theta(R^t, t)\|_2^2 \right] \tag{64}$$

$$= \arg\min_\theta \frac{\sigma_1^2}{2\sigma_2^2\mu_1^2} \left[ \|\epsilon_0 - \hat{\epsilon}_\theta(R^t, t)\|_2^2 \right] \tag{65}$$

Therefore, the training objective of the DPM can be written as

$$\mathcal{L}(\theta) = \mathbb{E}_{t,R^0,\epsilon}[\frac{\sigma_1^2}{2\mu_1^2\sigma_2^2}\|\epsilon - \epsilon_\theta(\mu_1\mu_2 R^0 + \sqrt{\mu_1^2\sigma_2^2 + \sigma_1^2}\epsilon, t)\|^2], \tag{66}$$

During the reverse process, we sample $R^{t-1} \sim p_\theta(R^{t-1}|R^t)$. Formally, the sampling (reverse) process is

$$R^{t-1} = \frac{1}{\mu_1} \left( R^t - \frac{\sigma_1^2}{\sqrt{\mu_1^2\sigma_2^2 + \sigma_1^2}}\epsilon_\theta(R^t, t) \right) + \frac{\sigma_1\sigma_2}{\sqrt{\mu_1^2\sigma_2^2 + \sigma_1^2}}z, \quad z \sim \mathcal{N}(\mathbf{0}, \mathbf{I}) \tag{67}$$

This lemma can be easily extended to the conditional version:

**Lemma C.2** *Assume the forward and reverse processes of the diffusion model are both Markov chains. $\{R^t\}_{t=0}^T$ are the states and $y_1$, $y_2$ are the given conditions. Given the forward Gaussian distribution*

$$q(R^t|R^{t-1}, R^0, y_1) = \mathcal{N}(R^t; \mu_1 R^{t-1}, \sigma_1^2 \boldsymbol{I});$$
$$q(R^{t-1}|R^0, y_2) = \mathcal{N}(R^{t-1}; \mu_2 R^0, \sigma_2^2 \boldsymbol{I})$$

*and $\epsilon_0 \sim \mathcal{N}(\mathbf{0}, \mathbf{I})$, we have the distribution $q(R^t|R^0, y_1, y_2) = \mathcal{N}(R^t; \mu_1 \mu_2 R^0, (\sigma_1^2 + \mu_1^2 \sigma_2^2)\boldsymbol{I})$. Thus, the posterior distribution $q(R^{t-1}|R^t, R^0)$ is*

$$q(R^{t-1}|R^t, R^0) \propto \mathcal{N}\left(R^{t-1}; \frac{1}{\mu_1}\left(R^t - \frac{\sigma_1^2}{\sqrt{\mu_1^2\sigma_2^2 + \sigma_1^2}}\epsilon_0\right), \frac{\sigma_1^2\sigma_2^2}{\mu_1^2\sigma_2^2 + \sigma_1^2}\boldsymbol{I}\right).$$

*Parameterizing $p_\theta(R^{t-1}|R^t, y_1, y_2)$ in the reverse process as*

$$\mathcal{N}\left(R^{t-1}; \frac{1}{\mu_1}\left(R^t - \frac{\sigma_1^2}{\sqrt{\mu_1^2\sigma_2^2 + \sigma_1^2}}\epsilon_\theta(R^t, t)\right), \frac{\sigma_1^2\sigma_2^2}{\mu_1^2\sigma_2^2 + \sigma_1^2}\boldsymbol{I}\right),$$

*the training objective of the DPM can be written as*

$$\mathcal{L}(\theta) = \mathbb{E}_{t, R^0, \epsilon}\left[\frac{\sigma_1^2}{2\mu_1^2\sigma_2^2}\|\epsilon - \epsilon_\theta(\mu_1\mu_2 R^0 + \sqrt{\mu_1^2\sigma_2^2 + \sigma_1^2}\epsilon, t)\|^2\right],$$

*and the sampling process is*

$$R^{t-1} = \frac{1}{\mu_1}\left(R^t - \frac{\sigma_1^2}{\sqrt{\mu_1^2\sigma_2^2 + \sigma_1^2}}\epsilon_\theta(R^t, t)\right) + \frac{\sigma_1\sigma_2}{\sqrt{\mu_1^2\sigma_2^2 + \sigma_1^2}}z, \tag{68}$$

*where $z \sim \mathcal{N}(\mathbf{0}, \mathbf{I})$.*

# D   Derivations of training objectives

## D.1   SubgDiff (1-same step and without expectation state)

Here, we utilize the binary characteristic of the mask vector to derive the ELBO for SubgDiff, and we also provide a general proof in Section D.2:

$$\log p(R^0) \geq \mathbb{E}_{q(R^{1:T}, s_{1:T}|R^0)}\left[\log \frac{p(R^{0:T}, s_{1:T})}{q(R^{1:T}|R^0, s_{1:T})q(s_{1:T})}\right] \tag{69}$$

$$= \mathbb{E}_{q(R^{1:T}, s_{1:T}|R^0)}\left[\log \frac{p(R^T)\prod_{t=1}^T p_\theta(R^{t-1}, s_t|R^t)}{\prod_{t=1}^T q(R^t|R^{t-1}, s_t)q(s_t)}\right] \tag{70}$$

$$= \mathbb{E}_{q(R^{1:T}, s_{1:T}|R^0)}\left[\log \frac{p(R^T)\prod_{t=1}^T p_\theta(R^{t-1}|R^t)p_\theta(s_t|R^t)}{\prod_{t=1}^T q(R^t|R^{t-1}, s_t)q(s_t)}\right] \tag{71}$$

$$= \mathbb{E}_{q(R^{1:T}, s_{1:T}|R^0)}\left[\log \frac{\prod_{t=1}^T p_\theta(s_t|R^t)}{\prod_{t=1}^T q(s_t)} + \log \frac{p(R^T)\prod_{t=1}^T p_\theta(R^{t-1}|R^t)}{\prod_{t=1}^T q(R^t|R^{t-1}, s_t)}\right] \tag{72}$$

$$= \underbrace{\mathbb{E}_{q(R^{1:T}, s_{1:T}|R^0)}\left[\sum_{t=1}^T \log \frac{p_\theta(s_t|R^t)}{q(s_t)}\right]}_{\text{mask prediction term}} + \mathbb{E}_{q(R^{1:T}, s_{1:T}|R^0)}\left[\log \frac{p(R^T)\prod_{t=1}^T p_\theta(R^{t-1}|R^t)}{\prod_{t=1}^T q(R^t|R^{t-1}, s_t)}\right]$$

$$\tag{73}$$
$$\tag{74}$$

The first term is mask prediction while the second term is similar to the ELBO of the classical diffusion model. The only difference is the $s_t$ in $q(R^t|R^{t-1}, s_t)$. According to Bayes rule, we can rewrite each transition as:

$$q(R^t|R^{t-1}, R^0, s_t) = \begin{cases} \frac{q(R^{t-1}|R^t, R^0)q(R^t|R^0)}{q(R^{t-1}|R^0)}, & \text{if } s_t = 1 \\ \delta_{R_{t-1}}(R_t). & \text{if } s_t = 0 \end{cases} \tag{75}$$

where $\delta_a(x) := \delta(x - a)$ is Dirac delta function, that is, $\delta_a(x) = 0$ if $x \neq a$ and $\int_{-\infty}^{\infty} \delta_a(x)dx = 1$. Without loss of generality, assume that $s_1$ and $s_T$ both equal 1. Armed with this new equation, we drive the second term:

$$\mathbb{E}_{q(R^{1:T}, s_{1:T}|R^0)}\left[\log \frac{p(R^T)\prod_{t=1}^{T} p_{\boldsymbol{\theta}}(R^{t-1}|R^t)}{\prod_{t=1}^{T} q(R^t|R^{t-1}, s_t)}\right] \tag{76}$$

$$= \mathbb{E}_{q(R^{1:T}, s_{1:T}|R^0)}\left[\log \frac{p(R^T)p_{\boldsymbol{\theta}}(R^0|R^1)\prod_{t=2}^{T} p_{\boldsymbol{\theta}}(R^{t-1}|R^t)}{q(R^1|R^0)\prod_{t=2}^{T} q(R^t|R^{t-1}, s_t)}\right] \tag{77}$$

$$= \mathbb{E}_{q(R^{1:T}, s_{1:T}|R^0)}\left[\log \frac{p(R^T)p_{\boldsymbol{\theta}}(R^0|R^1)\prod_{t=2}^{T} p_{\boldsymbol{\theta}}(R^{t-1}|R^t)}{q(R^1|R^0)\prod_{t=2}^{T} q(R^t|R^{t-1}, R^0, s_t)}\right] \tag{78}$$

$$= \mathbb{E}_{q(R^{1:T}, s_{1:T}|R^0)}\left[\log \frac{p_{\boldsymbol{\theta}}(R^T)p_{\boldsymbol{\theta}}(R^0|R^1)}{q(R^1|R^0)} + \log \prod_{t=2}^{T} \frac{p_{\boldsymbol{\theta}}(R^{t-1}|R^t)}{q(R^t|R^{t-1}, R^0, s_t)}\right] \tag{79}$$

$$= \mathbb{E}_{q(R^{1:T}, s_{1:T}|R^0)}\left[\log \frac{p(R^T)p_{\boldsymbol{\theta}}(R^0|R^1)}{q(R^1|R^0)} + \log \prod_{t\in\{t|s_t=1\}} \frac{p_{\boldsymbol{\theta}}(R^{t-1}|R^t)}{\frac{q(R^{t-1}|R^t, R^0)q(R^t|R^0)}{q(R^{t-1}|R^0, s_1)}} + \log \prod_{t\in\{t|s_t=0\}} \frac{p_{\boldsymbol{\theta}}(R^{t-1}|R^t)}{\delta_{R^{t-1}}(R^t)}\right] \tag{80}$$

$$= \mathbb{E}_{q(R^{1:T}|R^0)}\left[\log \frac{p(R^T)p_{\boldsymbol{\theta}}(R^0|R^1)}{q(R^1|R^0)} + \log \prod_{t\in\{t|s_t=0\}} \frac{p_{\boldsymbol{\theta}}(R^{t-1}|R^t)}{\delta_{R^{t-1}}(R^t)} + \log \prod_{t\in\{t|s_t=1\}} \frac{p_{\boldsymbol{\theta}}(R^{t-1}|R^t)}{\frac{q(R^{t-1}|R^t, R^0)\cancel{q(R^t|R^0)}}{\cancel{q(R^{t-1}|R^0)}}}\right] \tag{81}$$

$$= \mathbb{E}_{q(R^{1:T}|R^0)}\left[\log \prod_{t\in\{t|s_t=0\}} \frac{p_{\boldsymbol{\theta}}(R^{t-1}|R^t)}{\delta_{R^{t-1}}(R^t)} + \log \frac{p(R^T)p_{\boldsymbol{\theta}}(R^0|R^1)}{\cancel{q(R^1|R^0)}} + \log \frac{\cancel{q(R^1|R^0)}}{q(R^T|R^0)} + \log \prod_{t\in\{t|s_t=1\}} \frac{p_{\boldsymbol{\theta}}(R^{t-1}|R^t)}{q(R^{t-1}|R^t, R^0)}\right] \tag{82}$$

$$= \mathbb{E}_{q(R^{1:T}|R^0)}\left[\sum_{t\in\{t|s_t=0\}} \log \frac{p_{\boldsymbol{\theta}}(R^{t-1}|R^t)}{\delta_{R^{t-1}}(R^t)} + \log \frac{p(R^T)p_{\boldsymbol{\theta}}(R^0|R^1)}{q(R^T|R^0)} + \sum_{t\in\{t|s_t=1\}} \log \frac{p_{\boldsymbol{\theta}}(R^{t-1}|R^t)}{q(R^{t-1}|R^t, R^0)}\right] \tag{83}$$

$$= \sum_{t\in\{t|s_t=0\}} \mathbb{E}_{q(R^{1:T}|R^0)}\left[\log \frac{p_{\boldsymbol{\theta}}(R^{t-1}|R^t)}{\delta_{R^{t-1}}(R^t)}\right] + \mathbb{E}_{q(R^{1:T}|R^0)}\left[\log p_{\boldsymbol{\theta}}(R^0|R^1)\right] \tag{84}$$

$$+ \mathbb{E}_{q(R^{1:T}|R^0)}\left[\log \frac{p(R^T)}{q(R^T|R^0)}\right] + \sum_{t\in\{t|s_t=1\}} \mathbb{E}_{q(R^{1:T}|R^0)}\left[\log \frac{p_{\boldsymbol{\theta}}(R^{t-1}|R^t)}{q(R^{t-1}|R^t, R^0)}\right] \tag{85}$$

$$= \sum_{t\in\{t|s_t=0\}} \mathbb{E}_{q(R^{1:T}|R^0)}\left[\log \frac{p_{\boldsymbol{\theta}}(R^{t-1}|R^t)}{\delta_{R^{t-1}}(R^t)}\right] + \mathbb{E}_{q(R^1|R^0)}\left[\log p_{\boldsymbol{\theta}}(R^0|R^1)\right] \tag{86}$$

$$+ \mathbb{E}_{q(R^T|R^0)}\left[\log \frac{p(R^T)}{q(R^T|R^0)}\right] + \sum_{t\in\{t|s_t=1\}} \mathbb{E}_{q(R^t, R^{t-1}|R^0)}\left[\log \frac{p_{\boldsymbol{\theta}}(R^{t-1}|R^t)}{q(R^{t-1}|R^t, R^0)}\right] \tag{87}$$

$$= \underbrace{\sum_{t\in\{t|s_t=0\}} \mathbb{E}_{q(R^{1:T}|R^0)}\left[\log \frac{p_{\boldsymbol{\theta}}(R^{t-1}|R^t)}{\delta_{R^{t-1}}(R^t)}\right]}_{\textbf{decay term}} + \underbrace{\mathbb{E}_{q(R^1|R^0)}\left[\log p_{\boldsymbol{\theta}}(R^0|R^1)\right]}_{\text{reconstruction term}} \tag{88}$$

$$- \underbrace{D_{\text{KL}}(q(R^T|R^0) \parallel p(R^T))}_{\text{prior matching term}} - \sum_{t\in\{t|s_t=1\}} \underbrace{\mathbb{E}_{q(R^t|R^0)}\left[D_{\text{KL}}(q(R^{t-1}|R^t, R^0) \parallel p_{\boldsymbol{\theta}}(R^{t-1}|R^t))\right]}_{\text{denoising matching term}} \tag{89}$$

Here, the *decay term* represents the terms with $s_t = 0$, which are unnecessary to minimize when we set $p_{\boldsymbol{\theta}}(R^{t-1}|R^t) := \delta_{R^{t-1}}(R^t)$. Eventually, the ELOB can be rewritten as follows:

$$\log p(R^0) \geq \sum_{t=1}^{T} \underbrace{\mathbb{E}_{q(R^{1:T}|R^0)}\left[\log \frac{p_{\vartheta}(s_t|R^t)}{q(s_t)}\right]}_{\text{mask prediction term}} + \underbrace{\mathbb{E}_{q(R^1|R^0)}\left[\log p_{\boldsymbol{\theta}}(R^0|R^1)\right]}_{\text{reconstruction term}}$$

$$\underbrace{- D_{\text{KL}}(q(R^T|R^0) \parallel p(R^T))}_{\text{prior matching term}} - \sum_{t \in \{t|s_t=1\}} \underbrace{\mathbb{E}_{q(R^t|R^0)}\left[D_{\text{KL}}(q(R^{t-1}|R^t,R^0) \parallel p_{\boldsymbol{\theta}}(R^{t-1}|R^t))\right]}_{\text{denoising matching term}}$$

$$\tag{90}$$

The mask prediction term can be implemented by a node classifier and the denoising matching term can be calculated via Lemma C.1. In detail,

$$q(R^t|R^{t-1}, R^0) = \mathcal{N}(R^{t-1}, \sqrt{1-\beta_t s_t} R^{t-1}, (\beta_t s_t)\mathbf{I}), \tag{91}$$

$$q(R^{t-1}|R^0) = \mathcal{N}(R^{t-1}, \sqrt{\bar{\gamma}_{t-1}} R^0, (1-\bar{\gamma}_{t-1})\mathbf{I}). \tag{92}$$

Thus, the training objective of SubgDiff is:

$$\mathcal{L}(\theta, \vartheta) = \mathbb{E}_{t,R^0,\epsilon}\left[\frac{s_t\beta_t}{2(1-s_t\beta_t)(1-\bar{\gamma}_{t-1})}\|\epsilon - \epsilon_{\theta}(\sqrt{\bar{\gamma}_t}R^0 + \sqrt{(1-\bar{\gamma}_t)}\epsilon, t, \mathcal{G})\|^2 + \lambda \text{BCE}(\mathbf{s}_t, s_{\vartheta}(\mathcal{G}, \mathbf{R}^t, t))\right]$$

$$\tag{93}$$

To recover the existing work, we omit the mask prediction term (i.e. Let $p_{\theta}(s_t|R^t) := q(s_t)$) of SubgDiff in the main text.

### D.2   ELBO of SubgDiff

Here, we can derive the ELBO for SubgDiff:

$$\log p(R^0) = \log \int \int p(R^{0:T}, s_{1:T}) dR^{1:T} ds_{1:T} \tag{94}$$

$$= \log \int \int \frac{p(R^{0:T}, s_{1:T})q(R^{1:T}, s_{1:T}|R^0)}{q(R^{1:T}, s_{1:T}|R^0)} dR^{1:T} ds_{1:T} \tag{95}$$

$$= \log \int \int \left[\frac{p(R^{0:T}, s_{1:T})q(R^{1:T}|R^0, s_{1:T})q(s_{1:T})}{q(R^{1:T}, s_{1:T}|R^0)}\right] dR^{1:T} ds_{1:T} \tag{96}$$

$$= \log \mathbb{E}_{q(s_{1:T})}\mathbb{E}_{q(R^{1:T}|R^0, s_{1:T})}\left[\frac{p(R^{0:T}, s_{1:T}))}{q(R^{1:T}, s_{1:T}|R^0)}\right] \tag{97}$$

$$\geq \mathbb{E}_{q(R^{1:T}|R^0, s_{1:T})}\left[\log \mathbb{E}_{q(s_{1:T})}\frac{p(R^{0:T}, s_{1:T})}{q(R^{1:T}|R^0, s_{1:T})q(s_{1:T})}\right] \tag{98}$$

$$\geq \mathbb{E}_{q(R^{1:T}, s_{1:T}|R^0)}\left[\log \frac{p(R^T)\prod_{t=1}^{T} p_{\boldsymbol{\theta}}(R^{t-1}, s_t|R^t)}{\prod_{t=1}^{T} q(R^t|R^{t-1}, s_t)q(s_t)}\right] \tag{99}$$

$$= \mathbb{E}_{q(R^{1:T}, s_{1:T}|R^0)}\left[\log \frac{p(R^T)\prod_{t=1}^{T} p_{\boldsymbol{\theta}}(R^{t-1}|R^t)p_{\boldsymbol{\theta}}(s_t|R^t)}{\prod_{t=1}^{T} q(R^t|R^{t-1}, s_t)q(s_t)}\right] \tag{100}$$

$$= \mathbb{E}_{q(R^{1:T}, s_{1:T}|R^0)}\left[\log \frac{\prod_{t=1}^{T} p_{\theta}(s_t|R^t)}{\prod_{t=1}^{T} q(s_t)} + \log \frac{p(R^T)\prod_{t=1}^{T} p_{\boldsymbol{\theta}}(R^{t-1}|R^t, s_t)}{\prod_{t=1}^{T} q(R^t|R^{t-1}, s_t)}\right] \tag{101}$$

$$= \underbrace{\mathbb{E}_{q(R^{1:T}, s_{1:T}|R^0)}\left[\sum_{t=1}^{T} \log \frac{p_{\theta}(s_t|R^t)}{q(s_t)}\right]}_{\text{mask prediction term}} + \mathbb{E}_{q(R^{1:T}, s_{1:T}|R^0)}\left[\log \frac{p(R^T)\prod_{t=1}^{T} p_{\boldsymbol{\theta}}(R^{t-1}|R^t, s_t)}{\prod_{t=1}^{T} q(R^t|R^{t-1}, s_t)}\right]$$

$$\tag{102}$$

$$\tag{103}$$

According to Bayes rule, we can rewrite each transition as:

$$q(R^t|R^{t-1}, R^0, s_{1:t}) = \frac{q(R^{t-1}|R^t, R^0, s_{1:t})q(R^t|R^0, s_{1:t})}{q(R^{t-1}|R^0, s_{1:t-1})}, \tag{104}$$

Armed with this new equation, we drive the second term:

$$\mathbb{E}_{q(R^{1:T},s_{1:T}|R^0)}\left[\log\frac{p(R^T)\prod_{t=1}^T p_{\boldsymbol{\theta}}(R^{t-1}|R^t,s_t)}{\prod_{t=1}^T q(R^t|R^{t-1},s_t)}\right] \tag{105}$$

$$=\mathbb{E}_{q(R^{1:T},s_{1:T}|R^0)}\left[\log\frac{p(R^T)p_{\boldsymbol{\theta}}(R^0|R^1,s_1)\prod_{t=2}^T p_{\boldsymbol{\theta}}(R^{t-1}|R^t,s_t)}{q(R^1|R^0,s_1)\prod_{t=2}^T q(R^t|R^{t-1},s_t)}\right] \tag{106}$$

$$=\mathbb{E}_{q(R^{1:T},s_{1:T}|R^0)}\left[\log\frac{p(R^T)p_{\boldsymbol{\theta}}(R^0|R^1,s_1)\prod_{t=2}^T p_{\boldsymbol{\theta}}(R^{t-1}|R^t,s_t)}{q(R^1|R^0,s_1)\prod_{t=2}^T q(R^t|R^{t-1},R^0,s_{1:t})}\right] \tag{107}$$

$$=\mathbb{E}_{q(R^{1:T},s_{1:T}|R^0)}\left[\log\frac{p(R^T)p_{\boldsymbol{\theta}}(R^0|R^1,s_1)}{q(R^1|R^0,s_1)}+\log\prod_{t=2}^T\frac{p_{\boldsymbol{\theta}}(R^{t-1}|R^t,s_t)}{q(R^t|R^{t-1},R^0,s_{1:t})}\right] \tag{108}$$

$$=\mathbb{E}_{q(R^{1:T},s_{1:T}|R^0)}\left[\log\frac{p(R^T)p_{\boldsymbol{\theta}}(R^0|R^1,s_1)}{q(R^1|R^0,s_1)}+\log\prod_{t=2}^T\frac{p_{\boldsymbol{\theta}}(R^{t-1}|R^t,s_t)}{\frac{q(R^{t-1}|R^t,R^0,s_{1:t})q(R^t|R^0,s_{1:t})}{q(R^{t-1}|R^0,s_{1:t-1})}}\right] \tag{109}$$

$$=\mathbb{E}_{q(R^{1:T},s_{1:t}|R^0)}\left[\log\frac{p(R^T)p_{\boldsymbol{\theta}}(R^0|R^1,s_1)}{q(R^1|R^0,s_1)}+\log\prod_{t=2}^T\frac{p_{\boldsymbol{\theta}}(R^{t-1}|R^t,s_t)}{\frac{q(R^{t-1}|R^t,R^0,s_{1:t})\cancel{q(R^t|R^0,s_{1:t})}}{\cancel{q(R^{t-1}|R^0,s_{1:t-1})}}}\right] \tag{110}$$

$$=\mathbb{E}_{q(R^{1:T},s_{1:t}|R^0)}\left[\log\frac{p(R^T)p_{\boldsymbol{\theta}}(R^0|R^1,s_1)}{\cancel{q(R^1|R^0,s_1)}}+\log\frac{\cancel{q(R^1|R^0,s_1)}}{q(R^T|R^0,s_{1:T})}+\log\prod_{t=2}^T\frac{p_{\boldsymbol{\theta}}(R^{t-1}|R^t,s_t)}{q(R^{t-1}|R^t,R^0,s_{1:t})}\right] \tag{111}$$

$$=\mathbb{E}_{q(R^{1:T},s_{1:t}|R^0)}\left[\log\frac{p(R^T)p_{\boldsymbol{\theta}}(R^0|R^1,s_1)}{q(R^T|R^0,s_{1:T})}+\sum_{t=2}^T\log\frac{p_{\boldsymbol{\theta}}(R^{t-1}|R^t,s_t)}{q(R^{t-1}|R^t,R^0,s_{1:t})}\right] \tag{112}$$

$$=\mathbb{E}_{q(R^{1:T},s_{1:t}|R^0)}\left[\log p_{\boldsymbol{\theta}}(R^0|R^1,s_1)\right] \tag{113}$$

$$+\mathbb{E}_{q(R^{1:T},s_{1:t}|R^0)}\left[\log\frac{p(R^T)}{q(R^T|R^0,s_{1:T})}\right]+\sum_{t=2}^T\mathbb{E}_{q(R^{1:T},s_{1:t}|R^0)}\left[\log\frac{p_{\boldsymbol{\theta}}(R^{t-1}|R^t,s_t)}{q(R^{t-1}|R^t,R^0,s_{1:t})}\right] \tag{114}$$

$$=\mathbb{E}_{q(R^1,s_1|R^0)}\left[\log p_{\boldsymbol{\theta}}(R^0|R^1,s_1)\right] \tag{115}$$

$$+\mathbb{E}_{q(R^T|R^0,s_{1:T})q(s_{1:T})}\left[\log\frac{p(R^T)}{q(R^T|R^0,s_{1:T})}\right]+\sum_{t=2}^T\mathbb{E}_{q(R^t,R^{t-1},s_{1:t}|R^0)}\left[\log\frac{p_{\boldsymbol{\theta}}(R^{t-1}|R^t,s_t)}{q(R^{t-1}|R^t,R^0,s_{1:t})}\right] \tag{116}$$

$$=\underbrace{\mathbb{E}_{q(R^1,s_1|R^0)}\left[\log p_{\boldsymbol{\theta}}(R^0|R^1,s_1)\right]}_{\text{reconstruction term}} \tag{117}$$

$$-\underbrace{\mathbb{E}_{q(s_{1:t})}D_{\text{KL}}(q(R^T|R^0,s_{1:T})\parallel p(R^T))}_{\text{prior matching term}}-\sum_{t=2}^T\underbrace{\mathbb{E}_{q(R^t,s_{1:t}|R^0)}\left[D_{\text{KL}}(q(R^{t-1}|R^t,R^0,s_{1:t})\parallel p_{\boldsymbol{\theta}}(R^{t-1}|R^t,s_t))\right]}_{\text{denoising matching term}} \tag{118}$$

Eventually, the ELOB can be rewritten as follows:

$$\log p(R^0)\geq\sum_{t=1}^T\underbrace{\mathbb{E}_{q(R^t,s_t|R^0)}\left[\log\frac{p_{\vartheta}(s_t|R^t)}{q(s_t)}\right]}_{\text{mask prediction term}}+\underbrace{\mathbb{E}_{q(R^1,s_1|R^0)}\left[\log p_{\boldsymbol{\theta}}(R^0|R^1,s_1)\right]}_{\text{reconstruction term}} \tag{119}$$

$$-\underbrace{\mathbb{E}_{q(s_{1:t})}D_{\text{KL}}(q(R^T|R^0,s_{1:T})\parallel p(R^T))}_{\text{prior matching term}}-\sum_{t=2}^T\underbrace{\mathbb{E}_{q(R^t,s_{1:t}|R^0)}\left[D_{\text{KL}}(q(R^{t-1}|R^t,R^0,s_{1:t})\parallel p_{\boldsymbol{\theta}}(R^{t-1}|R^t,s_t))\right]}_{\text{denoising matching term}} \tag{120}$$

The mask prediction term can be implemented by a node classifier $s_{\vartheta}$. For the denoising matching term, by Bayes rule, the $q(R^{t-1}|R^t,R^0,s_{1:t})$ can be written as:

$$q(R^{t-1}|R^t,R^0,s_{1:t})=\frac{q(R^t|R^{t-1},R^0,s_{1:t})q(R^{t-1}|R^0,s_{1:t-1})}{q(R^t|R^0,s_{1:t})}, \tag{121}$$

For the naive SubgDiff, we have

$$q(R^t|R^{t-1}, R^0, s_{1:t}) := \mathcal{N}(R^{t-1}, \sqrt{1 - \beta_t s_t} R^{t-1}, (\beta_t s_t)\mathbf{I}), \tag{122}$$

$$q(R^{t-1}|R^0, s_{1:t-1}) := \mathcal{N}(R^{t-1}, \sqrt{\bar{\gamma}_{t-1}} R^0, (1 - \bar{\gamma}_{t-1})\mathbf{I}). \tag{123}$$

Then the denoising matching term can also be calculated via Lemma C.1 (let $q(R^t|R^{t-1}, R^0) := q(R^t|R^{t-1}, R^0, s_{1:t})$, $q(R^{t-1}|R^0) := q(R^{t-1}|R^0, s_{1:t-1})$ and $p_\theta(R^{t-1}|R^t) = p_\theta(R^{t-1})$). Thus, the training objective of SubgDiff is:

$$\mathcal{L}(\theta, \vartheta) = \mathbb{E}_{t, R^0, \epsilon} \left[ \frac{s_t \beta_t}{2(1 - s_t \beta_t)(1 - \bar{\gamma}_{t-1})} \| \epsilon - \epsilon_\theta(\sqrt{\bar{\gamma}_t} R^0 + \sqrt{(1 - \bar{\gamma}_t)} \epsilon, t, \mathcal{G}) \|^2 + \lambda \mathrm{BCE}(\mathbf{s}_t, s_\vartheta(\mathcal{G}, \mathbf{R}^t, t)) \right] \tag{124}$$

### D.2.1 Expectation of $s_{1:T}$

The denoising matching term in (120) can be calculated by only sampling $(R^t, s_t)$ instead of $(R^t, s_{1:t})$. Specifically, we substitute (121) into the denoising matching term:

$$\mathbb{E}_{q(R^t, R^{t-1}, s_{1:t}|R^0)} \left[ \log \frac{p_\theta(R^{t-1}|R^t, s_t)}{q(R^{t-1}|R^t, R^0, s_{1:t})} \right] \tag{125}$$

$$= \mathbb{E}_{q(R^t, R^{t-1}, s_{1:t}|R^0)} \left[ \log \frac{p_\theta(R^{t-1}|R^t, s_t)}{\frac{q(R^t|R^{t-1}, R^0, s_{1:t}) q(R^{t-1}|R^0, s_{1:t-1})}{q(R^t|R^0, s_{1:t})}} \right] \tag{126}$$

$$= \mathbb{E}_{q(R^t, R^{t-1}, s_{1:t}|R^0)} \left[ \log \frac{p_\theta(R^{t-1}|R^t, s_t)}{\frac{q(R^t|R^{t-1}, R^0, s_t)}{q(R^t|R^0, s_{1:t})}} - \log q(R^{t-1}|R^0, s_{1:t-1}) \right] \tag{127}$$

$$\geq \mathbb{E}_{q(R^t, R^{t-1}, |R^0, s_{1:t})} \left[ \mathbb{E}_{q(s_{1:t})} \log p_\theta(R^{t-1}|R^t, s_t) q(R^t|R^0, s_{1:t}) \right] \tag{128}$$

$$- \mathbb{E}_{q(s_t)} \left[ \log \underbrace{\mathbb{E}_{q(s_{1:t-1})} q(R^{t-1}|R^0, s_{1:t-1})}_{:= q(\mathbb{E}_s R^{t-1}|R^0)} + \log \underbrace{\mathbb{E}_{q(s_{1:t-1})} q(R^t|R^{t-1}, R^0, s_{1:t})}_{:= q(R^t|\mathbb{E}_s R^{t-1}, R^0, s_t)} \right] \tag{129}$$

$$= \mathbb{E}_{q(R^t, R^{t-1}, |R^0, s_{1:t})} \left[ \mathbb{E}_{q(s_{1:t})} \log \frac{p_\theta(R^{t-1}|R^t, s_t)}{\frac{1}{q(R^t|R^0, s_{1:t})}} - \mathbb{E}_{q(s_t)} \log q(\mathbb{E}_s R^{t-1}|R^0) - \mathbb{E}_{q(s_t)} \log q(R^t|\mathbb{E}_s R^{t-1}, R^0, s_t) \right] \tag{130}$$

$$= \mathbb{E}_{q(R^t, R^{t-1}, |R^0, s_{1:t})} \left[ \mathbb{E}_{q(s_{1:t})} \log \frac{p_\theta(R^{t-1}|R^t, s_t)}{\frac{q(R^t|\mathbb{E}_s R^{t-1}, R^0, s_t) q(\mathbb{E}_s R^{t-1}|R^0)}{q(R^t|R^0, s_{1:t})}} \right] \tag{131}$$

$$= \mathbb{E}_{q(R^t, R^{t-1}, s_{1:t}|R^0)} \underbrace{\left[ \log \frac{p_\theta(R^{t-1}|R^t, s_t)}{\frac{q(R^t|\mathbb{E}_s R^{t-1}, R^0, s_t) q(\mathbb{E}_s R^{t-1}|R^0)}{q(R^t|R^0, s_{1:t})}} \right]}_{\text{denoising matching term}} \tag{132}$$

$$= \mathbb{E}_{q(R^t, R^{t-1}, s_{1:t}|R^0)} \left[ \log \frac{p_\theta(R^{t-1}|R^t, s_t)}{\hat{q}(R^{t-1}|R^t, R^0, s_{1:t})} \right] \tag{133}$$

$$= \underbrace{\mathbb{E}_{q(R^t, s_{1:t}|R^0)} \left[ D_{\mathrm{KL}}(\hat{q}(R^{t-1}|R^t, R^0, s_{1:t}) \parallel p_\theta(R^{t-1}|R^t, s_t)) \right]}_{\text{denoising matching term}} \tag{134}$$

Thus, we should focus on calculating the distribution

$$\hat{q}(R^{t-1}|R^t, R^0, s_{1:t}) := \frac{q(R^t|\mathbb{E}_s R^{t-1}, R^0, s_t) q(\mathbb{E}_s R^{t-1}|R^0)}{q(R^t|R^0, s_{1:t})} \tag{135}$$

By lemma C.1, if we can gain the expression of $q(R^t|\mathbb{E}_s R^{t-1}, R^0, s_t)$ and $q(\mathbb{E}_s R^{t-1}|R^0)$, we can get the training objective and sampling process.

### D.3 Single-step subgraph diffusion

#### D.3.1 Training

**I: Step** $0$ **to Step** $t - 1$ $(R^0 \rightarrow R^{t-1})$**:** The state space of the mask diffusion should be the mean of the random state.

$$\mathbb{E}_s R^t \sim \mathcal{N}(\mathbb{E}_s R^t; \sqrt{1 - \beta_t} \mathbb{E}_s R^{t-1}, \beta_t I) \tag{136}$$

$$q(R^t | R^0, s_{1:t}) = \mathcal{N}(R^t, \sqrt{\bar{\gamma}_t} R^0, (1 - \bar{\gamma}_t)\mathbf{I}). \tag{137}$$

Form ([137](#)), we have:

$$R^t = \sqrt{1 - s_t \beta_t} R^{t-1} + \sqrt{s_t \beta_t} \epsilon_{t-1} \tag{138}$$

$$\mathbb{E} R^t = (p\sqrt{1 - \beta_t} + 1 - p)\mathbb{E} R^{t-1} + p\sqrt{\beta_t} \epsilon_{t-1} \tag{139}$$

$$= (p\sqrt{1 - \beta_t} + 1 - p)(p\sqrt{1 - \beta_{t-1}} + 1 - p)\mathbb{E} R^{t-2} + (p\sqrt{1 - \beta_t} + 1 - p)p\sqrt{\beta_{t-1}}\epsilon_{t-2} + p\sqrt{\beta_t}\epsilon_{t-1} \tag{140}$$

$$= (p\sqrt{1 - \beta_t} + 1 - p)(p\sqrt{1 - \beta_{t-1}} + 1 - p)\mathbb{E} R^{t-2} + \sqrt{[(p\sqrt{1 - \beta_t} + 1 - p)p\sqrt{\beta_{t-1}}]^2 + [p\sqrt{\beta_t}]^2}\epsilon_{t-2} \tag{141}$$

$$= .... \tag{142}$$

$$= \prod_{i=1}^{t}(p\sqrt{1 - \beta_i} + 1 - p)R^0 + \sqrt{[\prod_{j=2}^{t}(p\sqrt{1 - \beta_j} + 1 - p)p\sqrt{\beta_1}]^2 + [\prod_{j=3}^{t}(p\sqrt{1 - \beta_j} + 1 - p)p\sqrt{\beta_2}]^2 + ... + \epsilon_0} \tag{143}$$

$$= \prod_{i=1}^{t}(p\sqrt{1 - \beta_i} + 1 - p)R^0 + \sqrt{\sum_{i=1}^{t}[\prod_{j=i+1}^{t}(p\sqrt{1 - \beta_j} + 1 - p)p\sqrt{\beta_i}]^2} \tag{144}$$

$$= \prod_{i=1}^{t}\sqrt{\alpha_i} R^0 + \sqrt{\sum_{i=1}^{t}[\prod_{j=i+1}^{t}\sqrt{\alpha_i}p\sqrt{\beta_i}]^2}\epsilon_0 \tag{145}$$

$$= \prod_{i=1}^{t}\sqrt{\alpha_i} R^0 + p\sqrt{\sum_{i=1}^{t}\prod_{j=i+1}^{t}\alpha_j \beta_i}\epsilon_0 \tag{146}$$

$$= \sqrt{\bar{\alpha}_t} R^0 + p\sqrt{\sum_{i=1}^{t}\frac{\bar{\alpha}_t}{\bar{\alpha}_i}\beta_i}\epsilon_0 \tag{147}$$

$$\tag{148}$$

where $\alpha_i := (p\sqrt{1 - \beta_i} + 1 - p)^2$ and $\bar{\alpha}_t = \prod_{i=1}^{t}\alpha_i$.

$$q(\mathbb{E} R^t | R^0) = \mathcal{N}(R^t; \sqrt{\bar{\alpha}_t} R^0, p^2 \sum_{i=1}^{t}\frac{\bar{\alpha}_t}{\bar{\alpha}_i}\beta_i I) \tag{149}$$

**II: Step $t - 1$ to Step $t$ ($R^{t-1} \to R^t$):** We build the step $t - 1 \to t$ is a discrete transition from $q(\mathbf{R}^{t-1} | \mathbf{R}^0)$, with

$$q(\mathbb{E}_s R^{t-1} | R^0) = \mathcal{N}(R^{t-1}; \prod_{i=1}^{t-1}\sqrt{\alpha_i} R^0, p^2 \sum_{i=1}^{t-1}\prod_{j=i+1}^{t-1}\alpha_j \beta_i I) \tag{150}$$

$$q(R^t | \mathbb{E}_s R^{t-1}, s_t) = \mathcal{N}(R^t; \sqrt{1 - s_t \beta_t}\mathbb{E} R^{t-1}, s_t \beta_t I) \tag{151}$$

$$R^t = \sqrt{1 - s_t\beta_t}\mathbb{E}R^{t-1} + \sqrt{s_t\beta_t}\epsilon_{t-1} \tag{152}$$

$$= \sqrt{1 - s_t\beta_t}\left(\sqrt{\bar{\alpha}_{t-1}}R^0 + p\sqrt{\sum_{i=1}^{t-1}\frac{\bar{\alpha}_{t-1}}{\bar{\alpha}_i}\beta_i\epsilon_0}\right) + \sqrt{s_t\beta_t}\epsilon_{t-1} \tag{153}$$

$$= \sqrt{1 - s_t\beta_t}\sqrt{\bar{\alpha}_{t-1}}R^0 + p\sqrt{1 - s_t\beta_t}\sqrt{\sum_{i=1}^{t-1}\frac{\bar{\alpha}_{t-1}}{\bar{\alpha}_i}\beta_i\epsilon_0} + \sqrt{s_t\beta_t}\epsilon_{t-1} \tag{154}$$

$$= \sqrt{1 - s_t\beta_t}\sqrt{\bar{\alpha}_{t-1}}R^0 + \sqrt{p^2(1 - s_t\beta_t)\sum_{i=1}^{t-1}\frac{\bar{\alpha}_{t-1}}{\bar{\alpha}_i}\beta_i + s_t\beta_t\epsilon_0} \tag{155}$$

**Step $0$ to Step $t$ ($R^0 \to R^t$):**

$$q(R^t|R^0) = \int q(R^t|\mathbb{E}R^{t-1})q(\mathbb{E}R^{t-1}|R^0)d\mathbb{E}R^{t-1} \tag{156}$$

$$= \mathcal{N}(R^t; \sqrt{1 - s_t\beta_t}\sqrt{\bar{\alpha}_i}R^0, (p^2(1 - s_t\beta_t)\sum_{i=1}^{t-1}\frac{\bar{\alpha}_{t-1}}{\bar{\alpha}_i}\beta_i + s_t\beta_t)I) \tag{157}$$

Thus, from subsection D.2.1, the **training objective** of 1-step SubgDiff is:

$$\mathcal{L}_{simple}(\theta, \vartheta) = \mathbb{E}_{t,R^0,s_t,\epsilon}[s_t\|\epsilon - \epsilon_\theta(R^t, t)\|^2 - \mathcal{BCE}(s_t, s_\vartheta(R^t, t))] \tag{158}$$

where $\mathcal{BCE}(s_t, s_\vartheta) = s_t \log s_\vartheta(R^t, t) + (1 - s_t)\log(1 - s_\vartheta(R^t, t))$ is Binary Cross Entropy loss. However, training the SubgDiff is not trivial. The challenges come from two aspects: 1) the mask predictor should be capable of perceiving the sensible noise change between $(t-1)$-th and $t$-th step. However, the noise scale $\beta_t$ is relatively small when $t$ is small, especially if the diffusion step is larger than a thousand, thereby mask predictor cannot precisely predict. 2) The accumulated noise for each node at $(t-1)$-th step would be mainly affected by the mask sampling from 1 to $t-1$ step, which heavily increases the difficulty of predicting the noise added between $(t-1)$-step to $t$-step.

### D.3.2 Sampling

Finally, the sampling can be written as:

$$R^{t-1} = \frac{\left((1 - s_t\beta_t)p^2\sum_{i=1}^{t-1}\frac{\bar{\alpha}_{t-1}}{\bar{\alpha}_i}\beta_i + s_t\beta_t\right)R^t - \left(s_t\beta_t\sqrt{p^2(1 - s_t\beta_t)\sum_{i=1}^{t-1}\frac{\bar{\alpha}_{t-1}}{\bar{\alpha}_i}\beta_i + s_t\beta_t}\right)\epsilon_\theta(R^t, t)}{\sqrt{1 - s_t\beta_t}(s_t\beta_t + (1 - s_t\beta_t)p^2\sum_{i=1}^{t-1}\frac{\bar{\alpha}_{t-1}}{\bar{\alpha}_i}\beta_i)} + \sigma_t z \tag{159}$$

$$= \frac{1}{\sqrt{1 - s_t\beta_t}}R^t - \frac{\left(s_t\beta_t\sqrt{p^2(1 - s_t\beta_t)\sum_{i=1}^{t-1}\frac{\bar{\alpha}_{t-1}}{\bar{\alpha}_i}\beta_i + s_t\beta_t}\right)}{\sqrt{1 - s_t\beta_t}(s_t\beta_t + (1 - s_t\beta_t)p^2\sum_{i=1}^{t-1}\frac{\bar{\alpha}_{t-1}}{\bar{\alpha}_i}\beta_i)}\epsilon_\theta(R^t, t) + \sigma_t z \tag{160}$$

$$= \frac{1}{\sqrt{1 - s_t\beta_t}}R^t - \frac{s_t\beta_t}{\sqrt{1 - s_t\beta_t}\sqrt{s_t\beta_t + (1 - s_t\beta_t)p^2\sum_{i=1}^{t-1}\frac{\bar{\alpha}_{t-1}}{\bar{\alpha}_i}\beta_i}}\epsilon_\theta(R^t, t) + \sigma_t z \tag{161}$$

$$\tag{162}$$

where $s_t = s_\vartheta(R^t, t)$ and

$$\sigma_t = \frac{s_\vartheta(R^t, t)\beta_t p^2\sum_{i=1}^{t-1}\frac{\bar{\alpha}_{t-1}}{\bar{\alpha}_i}\beta_i}{s_\vartheta(R^t, t)\beta_t + p^2(1 - s_\vartheta(R^t, t)\beta_t)\sum_{i=1}^{t-1}\frac{\bar{\alpha}_{t-1}}{\bar{\alpha}_i}\beta_i} \tag{163}$$

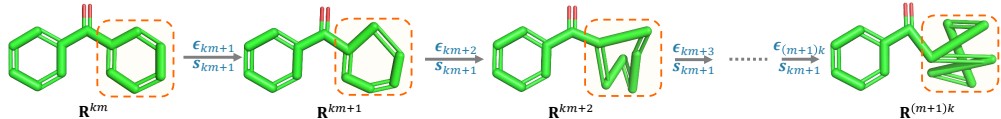

Figure 8: An example of $k$-step same subgraph diffusion, where the mask vectors are same as $\mathbf{s}_{km+1}$ from step $km$ to $(m+1)k$, $m \in \mathbb{N}^+$ .

## E    Expectation state distribution

The state space of the mask diffusion should be the mean of the random state.

$$\mathbb{E}_{s_t} R^t \sim \mathcal{N}(\mathbb{E}R^t; \sqrt{1-\beta_t}\mathbb{E}_{s_{t-1}}R^{t-1}, \beta_t I) \tag{164}$$

Form Equation 137, we have:

$$R^t = \sqrt{1-s_t\beta_t}R^{t-1} + \sqrt{s_t\beta_t}\epsilon_{t-1} \tag{165}$$

$$\mathbb{E}R^t = (p\sqrt{1-\beta_t}+1-p)\mathbb{E}R^{t-1} + p\sqrt{\beta_t}\epsilon_{t-1} \tag{166}$$

$$= (p\sqrt{1-\beta_t}+1-p)(p\sqrt{1-\beta_{t-1}}+1-p)\mathbb{E}R^{t-2} \tag{167}$$

$$+ (p\sqrt{1-\beta_t}+1-p)p\sqrt{\beta_{t-1}}\epsilon_{t-2} + p\sqrt{\beta_t}\epsilon_{t-1} \tag{168}$$

$$= (p\sqrt{1-\beta_t}+1-p)(p\sqrt{1-\beta_{t-1}}+1-p)\mathbb{E}R^{t-2} \tag{169}$$

$$+ \sqrt{[(p\sqrt{1-\beta_t}+1-p)p\sqrt{\beta_{t-1}}]^2 + [p\sqrt{\beta_t}]^2}\epsilon_{t-2} \tag{170}$$

$$= \dots. \tag{171}$$

$$= \prod_{i=1}^{t}(p\sqrt{1-\beta_i}+1-p)R^0 \tag{172}$$

$$+ \sqrt{[\prod_{j=2}^{t}(p\sqrt{1-\beta_j}+1-p)p\sqrt{\beta_1}]^2 + [\prod_{j=3}^{t}(p\sqrt{1-\beta_j}+1-p)p\sqrt{\beta_2}]^2 + \dots +\epsilon_0} \tag{173}$$

$$= \prod_{i=1}^{t}(p\sqrt{1-\beta_i}+1-p)R^0 + \sqrt{\sum_{i=1}^{t}[\prod_{j=i+1}^{t}(p\sqrt{1-\beta_j}+1-p)p\sqrt{\beta_i}]^2} \tag{174}$$

$$= \prod_{i=1}^{t}\sqrt{\alpha_i}R^0 + \sqrt{\sum_{i=1}^{t}[\prod_{j=i+1}^{t}\sqrt{\alpha_i}p\sqrt{\beta_i}]^2}\epsilon_0 \tag{175}$$

$$= \prod_{i=1}^{t}\sqrt{\alpha_i}R^0 + p\sqrt{\sum_{i=1}^{t}\prod_{j=i+1}^{t}\alpha_j\beta_i}\epsilon_0 \tag{176}$$

$$= \sqrt{\bar{\alpha}_i}R^0 + p\sqrt{\sum_{i=1}^{t}\frac{\bar{\alpha}_t}{\bar{\alpha}_i}\beta_i}\epsilon_0 \tag{177}$$

$$\tag{178}$$

where $\alpha_i := (p\sqrt{1-\beta_i}+1-p)^2$ and $\bar{\alpha}_t = \prod_{i=1}^{t}\alpha_i$.

Finally, the Expectation state distribution is:

$$q(\mathbb{E}R^t|R^0) = \mathcal{N}(\mathbb{E}R^t; \prod_{i=1}^{t}\sqrt{\alpha_i}R^0, p^2\sum_{i=1}^{t}\prod_{j=i+1}^{t}\alpha_j\beta_i I) \tag{179}$$

## F  The derivation of SubgDiff

When $t$ is an integer multiple of $k$,

$$\mathbb{E}R^t = \prod_{j=1}^{t/k}(p\sqrt{\prod_{i=(j-1)k+1}^{kj}(1-\beta_i)}+1-p)R^0 \tag{180}$$

$$+ \sqrt{\sum_{l=1}^{t/k}\left[\prod_{j=l+1}^{t/k}(p\sqrt{\prod_{i=(j-1)k+1}^{kj}(1-\beta_i)}+1-p)p\sqrt{1-\prod_{i=(l-1)k+1}^{kl}(1-\beta_i)}\right]^2}\epsilon_0 \tag{181}$$

$$= \prod_{j=1}^{t/k}\sqrt{\alpha_j}R^0 + p\sqrt{\sum_{l=1}^{t/k}\prod_{j=l+1}^{t/k}\alpha_j(1-\prod_{i=(l-1)k+1}^{kl}(1-\beta_i))}\epsilon_0 \tag{182}$$

$$= \sqrt{\bar{\alpha}_{t/k}}R^0 + p\sqrt{\sum_{l=1}^{t/k}\frac{\bar{\alpha}_{t/k}}{\bar{\alpha}_l}(1-\prod_{i=(l-1)k+1}^{kl}(1-\beta_i))}\epsilon_0 \tag{183}$$

where $\alpha_j = (p\sqrt{\prod_{i=(j-1)k+1}^{kj}(1-\beta_i)}+1-p)^2$.

When $t \in \mathbb{N}$, let $m := \lfloor(t-1)/k\rfloor$, and we have

$$R^t = \sqrt{\prod_{i=km+1}^{t}(1-\beta_i s_{km+1})}\mathbb{E}R^{m\times k} + \sqrt{1-\prod_{i=km+1}^{t}(1-\beta_i s_{km+1})}\epsilon_{m\times k} \tag{184}$$

$$= \sqrt{\prod_{i=km+1}^{t}(1-\beta_i s_{km+1})}\left(\sqrt{\bar{\alpha}_m}R^0 + p\sqrt{\sum_{l=1}^{m}\frac{\bar{\alpha}_m}{\bar{\alpha}_l}(1-\prod_{i=(l-1)k+1}^{kl}(1-\beta_i))}\epsilon_0\right) \tag{185}$$

$$+ \sqrt{1-\prod_{i=km+1}^{t}(1-\beta_i s_{km+1})}\epsilon_m \tag{186}$$

$$= \sqrt{\prod_{i=km+1}^{t}\gamma_i\sqrt{\bar{\alpha}_m}}R^0 \tag{187}$$

$$+ \sqrt{\left(\prod_{i=km+1}^{t}\gamma_i\right)p^2\sum_{l=1}^{m}\frac{\bar{\alpha}_m}{\bar{\alpha}_l}(1-\prod_{i=(l-1)k+1}^{kl}(1-\beta_i))+\left(1-\prod_{i=km+1}^{t}\gamma_i\right)}\epsilon_0 \tag{188}$$

where $\gamma_i = 1 - \beta_i s_{km+1}$.

$$q(R^t|R^0) = \mathcal{N}(R^{km}; \sqrt{\prod_{i=km+1}^{t}\gamma_i\sqrt{\bar{\alpha}_m}}R^0, \tag{189}$$

$$\left(\left(\prod_{i=km+1}^{t}\gamma_i\right)p^2\sum_{l=1}^{m}\frac{\bar{\alpha}_m}{\bar{\alpha}_l}(1-\prod_{i=(l-1)k+1}^{kl}(1-\beta_i))+1-\prod_{i=km+1}^{t}\gamma_i\right)I) \tag{190}$$

Let $\bar{\gamma}_i = \prod_{t=1}^{i}\gamma_t$, and $\bar{\beta}_t = \prod_{i=1}^{t}(1-\beta_i)$

$$q(R^t|R^0) = \mathcal{N}(R^{km}; \sqrt{\frac{\bar{\gamma}_t}{\bar{\gamma}_{km}}}\sqrt{\bar{\alpha}_m}R^0, \left(\frac{\bar{\gamma}_t}{\bar{\gamma}_{km}}p^2\sum_{l=1}^{m}\frac{\bar{\alpha}_m}{\bar{\alpha}_l}(1-\frac{\bar{\beta}_{kl}}{\bar{\beta}_{(l-1)k}})+1-\frac{\bar{\gamma}_t}{\bar{\gamma}_{km}}\right)I) \tag{191}$$

### F.0.1 Sampling

$$\mu_1 = \sqrt{1 - s_{km+1}\beta_t}, \tag{192}$$

$$\sigma_1^2 = s_{km+1}\beta_t \tag{193}$$

$$\mu_2 = \sqrt{\frac{\bar{\gamma}_{t-1}}{\bar{\gamma}_{km}}}\sqrt{\bar{\alpha}_m} \tag{194}$$

$$\sigma_2^2 = \frac{\bar{\gamma}_{t-1}}{\bar{\gamma}_{km}}p^2\sum_{l=1}^{m}\frac{\bar{\alpha}_m}{\bar{\alpha}_l}(1 - \prod_{i=(l-1)k+1}^{kl}(1-\beta_i)) + 1 - \frac{\bar{\gamma}_{t-1}}{\bar{\gamma}_{km}} \tag{195}$$

According to the Lemma C.1, we have

$$R^{t-1} = \frac{1}{\mu_1}\left(R^t - \frac{\sigma_1^2}{\sqrt{\mu_1^2\sigma_2^2 + \sigma_1^2}}\epsilon_\theta(R^t, t)\right) + \frac{\sigma_1\sigma_2}{\sqrt{\mu_1^2\sigma_2^2 + \sigma_1^2}}z \tag{196}$$

$$= \frac{1}{\sqrt{1 - s_{km+1}\beta_t}}(R^t - \tag{197}$$

$$\frac{s_{km+1}\beta_t}{\sqrt{(1 - s_{km+1}\beta_t)(\frac{\bar{\gamma}_{t-1}}{\bar{\gamma}_{km}}p^2\sum_{l=1}^{m}\frac{\bar{\alpha}_m}{\bar{\alpha}_l}(1 - \prod_{i=(l-1)k+1}^{kl}(1-\beta_i)) + 1 - \frac{\bar{\gamma}_{t-1}}{\bar{\gamma}_{km}}) + s_{km+1}\beta_t}}\epsilon_\theta(R^t, t)) \tag{198}$$

$$+ \frac{\sqrt{s_{km+1}\beta_t}\sqrt{\frac{\bar{\gamma}_{t-1}}{\bar{\gamma}_{km}}p^2\sum_{l=1}^{m}\frac{\bar{\alpha}_m}{\bar{\alpha}_l}(1 - \prod_{i=(l-1)k+1}^{kl}(1-\beta_i)) + 1 - \frac{\bar{\gamma}_{t-1}}{\bar{\gamma}_{km}}}}{\sqrt{(1 - s_{km+1}\beta_t)(\frac{\bar{\gamma}_{t-1}}{\bar{\gamma}_{km}}p^2\sum_{l=1}^{m}\frac{\bar{\alpha}_m}{\bar{\alpha}_l}(1 - \prod_{i=(l-1)k+1}^{kl}(1-\beta_i)) + 1 - \frac{\bar{\gamma}_{t-1}}{\bar{\gamma}_{km}}) + s_{km+1}\beta_t}}z \tag{199}$$

The schematic can see Figure 5.

