# OpenReview forum: "SubgDiff: A Subgraph Diffusion Model to Improve Molecular Representation Learning"
_NeurIPS.cc/2024/Conference — NeurIPS 2024 poster_

### Official Review · Reviewer_aMXv · 2024-07-07

**Soundness:** 2
**Presentation:** 2
**Contribution:** 3
**Rating:** 4
**Confidence:** 4

**Summary:**

SubgDiff  is introduced to improve molecular representation learning by integrating substructural information into the diffusion model framework. It offers three key technical contributions (subgraph prediction, expectation state, and k-step same subgraph diffusion) to enhance the network's understanding of molecular substructures. Experiments were carried out on several downstream tasks, particularly molecular force predictions.

**Strengths:**

* Incorporate substructural information into diffusion model

**Weaknesses:**

* The denoising process need better explanations.
* Diffusion models excel at generating new samples. The application of SubgDiff to molecular property prediction/classification does not show its strengths. The state-of-the-art results are missed in Section 5.1. Section 5.2 doesn't compare generation results with the state-of-the-art.
* Explain COV-R and MAT-R in section 5.2

**Questions:**

* Does the denoising processing require to start with a R^T that has a clear topology? If not, how to sample s_t?
* How to select k in the k-step same-subgraph diffusion? Any general guidance?
* How are the SubgDiff learning results fed into downstream tasks?

---

> ### Author Rebuttal · Authors · 2024-08-07
>
> We thank the reviewer for the constructive suggestions and useful feedback!
>
> > W1: The denoising process needs better explanations.
>
> **AW1:** Thanks for the suggestive advice. We will further describe the details of the denoising process in the paper. The only difference from the traditional diffusion backward process is that when $t\%k=0$, we use the subgraph predictor to update the subgraph to be denoised. Additionally, the whole denoising process can be referred to in Algorithm #1 below.
>
> > W2: Diffusion models excel at generating new samples. The application of SubgDiff to molecular property prediction/classification does not show its strengths. The state-of-the-art results are missing in Section 5.1. Section 5.2 doesn't compare generation results with the state-of-the-art.
>
> **AW2**: Thanks a lot for the insightful comments.
> We agree that diffusion models are good at generation. As a generation model, the objective of diffusion can be used as a task for self-supervised learning. This is one of our motivations and our primary focus is on improving the diffusion model for representation learning. Our experiments on molecular property prediction show that our approach outperforms various molecular self-supervised learning methods and the original diffusion model.
>
> (1) In Section 5.1, our method uses the same framework as MoleculeSDE, which is the state-of-the-art technique for molecular pretraining. The experiments also demonstrate our method can gain significant improvement over the other self-supervised techniques. Besides, our method significantly outperforms MoleculeSDE, especially in 3D property prediction (Table 1 in the main paper), which indicates the effectiveness of our method by incorporating the information of substructures.
>
> (2) In Section 5.2, we focus on evaluating the substructure's effect on the diffusion model itself and the experiment verifies the success of our design (Table 2 in the main text). Our results show that our approach is able to improve the sampling efficiency and generalization ability, compared to the original diffusion model. We also discuss other conformational generation models in the related work and provide the performance of other SOTA methods in Appendix Table 12.
>
> > W3: Explain COV-R and MAT-R in section 5.2
>
> **AW3:** Thanks for pointing this out. The definition of COV-R and MAT-R can be found in Eq (32) and Eq (33) in Appendix A.5.2.
>
> "Let $S_g$ and $S_r$ denote the sets of generated and reference conformers respectively, then the Coverage and Matching metrics  can be defined as
>
> $COV-R(S_g, S_r) = \frac{1}{ |S_r| }  |
>    \lbrace C\in S_r |  \operatorname{RMSD}(C, \hat{C}) \le \delta,  \hat{C} \in S_g \rbrace
> | $
>
> $ {MAT-R}(S_g, S_r) = \frac{1}{| S_r |}
>     \sum\limits_{C \in S_r}
>     \min\limits_{\hat{C} \in S_g} \operatorname{RMSD}(C, \hat{C})$
> where $\delta$ is a threshold."
>
> We will put it into Section 5.1 in the main paper.
>
>
> > Q1: Does the denoising processing require starting with a $R^T$ that has a clear topology? If not, how to sample s_t?
>
> **AQ1:** Thanks for the question. Our diffusion model is able to generate the molecule conformation conditioned on the molecule graph. So the graph topology is given. But the initial conformation $R^T$ is sampled from the Gaussian noise. And the $s_t$ is predicted through the subgraph predictor $s_\vartheta(\mathcal{G}, R^t,t)$ trained by the objective in Eq (17). Algorithm 2 in the main paper and Algorithm 4 in the appendix also describe the whole process. We also put the algorithm #1 below.
>
> - **Algorithm #1: Sampling from SubgDiff**
> ----
> Sample $R^T  \sim \mathcal N(\mathbf 0, \mathbf I)$ \//Random noise initialization
>
> **For**{t = T **To** 1}
>
> {
>    1.  $\mathbf z  \sim \mathcal N(\mathbf 0, \mathbf I)$ if $t>1$, else $\mathbf z =\mathbf 0$ \// Random noise
>
>   2. **If** $t\%k==0$  or $t==T$: $\hat{s} \gets s_\vartheta(\mathcal{G},R^{t},t)$ \//subgraph prediction
>
>    3.  $\hat\epsilon \gets \epsilon_\theta (\mathcal G,R^{t},t)$ \// Posterior
>
>    4.  $R^{t-1} \gets$ Equation 18 \// sampling coordinate
>
> }
>
> **Return** $R^0$
>
>
>
> >Q2: How to select k in the k-step same-subgraph diffusion? Any general guidance?
>
> **AQ2:** Thanks for this valuable question. $k$ cannot be too large. A large $k$ will make the subgraph prediction task too easy such that the model learns little substructure information. However, a very small $k$ makes it difficult for the subgraph predictor to converge. From our experience, $k$ and the number of the diffusion step $N$ can maintain a certain ratio, e.g. $N:k=20$ in our experiments, and the model can achieve better performance. In practice, $k$ is a hyper-parameter that can be tuned according to the sampling performance.
> We also provide the sensitivity analysis of $k$ in Table #1 of the general response.
>
>
> >Q3: How are the SubgDiff learning results fed into downstream tasks?
>
> **AQ3:** Thank you for the important question. The denoising network $\epsilon_\theta (\mathcal G, R^{t},t)$ contains a 3D encoder based on SchNet and a 2D encoder based on GIN. After we finish the training of SubgDiff, the encoders can be used as the pretrained models for the downstream tasks. This feeding method follows the conventional generative self-supervised techniques, e.g.MolecularSDE[1] and Denoising[2]. We also provide the details in Appendix A.4.
>
> [1] Shengchao Liu, et al. A group symmetric stochastic differential equation model for molecule multi-modal pretraining. ICML 2023
>
> [2] Sheheryar Zaidi, et al. Pre-training via denoising for molecular property prediction. ICLR 2023.

---

> > ### Author Response · Authors · 2024-08-12
> > **kind reminder to the reviewer**
> >
> > Dear reviewer,
> >
> > We kindly ask if you could inform us whether your concerns have been adequately addressed in our rebuttal, or if you have any further questions. We are committed to responding promptly to any additional inquiries you may have.
> >
> > Thank you for your time and valuable feedback.
> >
> > Best regards,
> >
> > The Authors

---

### Official Review · Reviewer_VBzc · 2024-07-12

**Soundness:** 2
**Presentation:** 2
**Contribution:** 3
**Rating:** 5
**Confidence:** 3

**Summary:**

The paper proposed SubgDiff which is a diffusion model used in self-supervised learning setup to enhance the molecular representation learning. It introduces motif enhancement during the diffusion process to force the model to learn more structure information.

**Strengths:**

1. The idea of enhancing motif information in the diffusion process is promising.
2. The paper is well-organized and easy to follow.

**Weaknesses:**

1. The authors directly use the baseline results from the MoleculeSDE paper in their table; however, the results for MoleculeSDE are significantly lower than those reported in the original paper.
2. The proposed method is more like a graph diffusion model than a molecular representation learning model. It is limited to representation learning within a self-supervised learning framework. It would be beneficial to explicitly state this in the abstract or introduction.
3. The paper lacks an ablation study to evaluate the contribution of each component of SubgDiff to molecular representation learning.

**Questions:**

Please refer to Weaknesses.

---

> ### Author Rebuttal · Authors · 2024-08-07
>
> Thank you for reviewing our paper and sharing these important points. These comments highlight several areas where we can improve the clarity and thoroughness of our paper. We acknowledge the issues raised and would like to address each point.
>
> > W1: The authors directly use the baseline results from the MoleculeSDE paper in their table; however, the results for MoleculeSDE are significantly lower than those reported in the original paper.
>
> **A1:** Thanks for your careful review! We use the released code provided by the MoleculeSDE and report the reproduced results. Our method applies the same pretraining framework as MoleculeSDE except for the diffusion model. Specifically, we replace the original diffusion model in MoleculeSDE with our SubgDiff. Hence, we used the reproduced results from MolecularSDE instead of the results reported in the paper for a fair comparison.
>
>
> > W2: The proposed method is more like a graph diffusion model than a molecular representation learning model. It is limited to representation learning within a self-supervised learning framework. It would be beneficial to explicitly state this in the abstract or introduction.
>
> **A2:** We agree with the reviewer that our method bears similarities to graph diffusion models and that its primary focus is on representation learning within a self-supervised framework. Indeed, we use the denoising objective in diffusion as the self-supervised task. We acknowledge that we should have been more explicit about this positioning of our work. To address this:
> a) We will revise the abstract to clearly state that SubgDiff is a graph diffusion model focused on molecular representation learning in a self-supervised context.
> b) In the introduction, we will add a paragraph explicitly discussing the nature of our model and its place within the broader landscape of molecular self-supervised learning.
> c) We will review the entire manuscript to ensure consistent and clear communication about the scope and nature of our method.
>
> > W3: The paper lacks an ablation study to evaluate the contribution of each component of SubgDiff to molecular representation learning.
>
> **A3:** Thank you for the helpful suggestion. (1) Our method contains three components, where the subgraph prediction and expectation state are designed to work together for the sampling. Without these two techniques, the method cannot be used for sampling due to the inaccessible $s_t$ during inference. For the $k$-step same subgraph component, we provide the results concerning different $k$ to demonstrate its significance, as shown in **Table #1** in the general response.
> (2) The expectation state and subgraph prediction can be evaluated in the self-supervised learning context. The results of the downstream force prediction tasks are shown in **Table #2** in the general response.
> The results suggest that the subgraph prediction loss plays a more important role in molecular representation learning.

---

> > ### Author Response · Authors · 2024-08-12
> > **Kind reminder to the reviewer**
> >
> > Dear reviewer,
> >
> > We kindly ask if you could inform us whether your concerns have been adequately addressed in our rebuttal, or if you have any further questions. We are committed to responding promptly to any additional inquiries you may have.
> >
> > Thank you for your time and valuable feedback.
> >
> > Best regards,
> >
> > The Authors

---

### Official Review · Reviewer_Djew · 2024-07-12

**Soundness:** 3
**Presentation:** 3
**Contribution:** 3
**Rating:** 6
**Confidence:** 4

**Summary:**

The paper presents a new denoising diffusion probabilistic model (DDPM) named SubgDiff, designed to enhance molecular representation learning by incorporating substructural information into the diffusion process. SubgDiff introduces a mask operation that selects subgraphs for diffusion, aiming to better capture the dependencies among atoms within substructures. The method includes techniques such as subgraph prediction, expectation state diffusion, and k-step same-subgraph diffusion, which together are intended to improve the learning of molecular properties related to 3D conformation. The paper claims superior performance on various downstream tasks, particularly in molecular force predictions.

**Strengths:**

1.	Utilizing subgraphs for diffusion is a novel and intriguing exploration.
2.	The experiments are thorough and demonstrate the effectiveness of SubgDiff across a range of molecular prediction tasks.

**Weaknesses:**

1.	The subgraph prediction model is trained on highly specialized datasets, which might pose a risk of overfitting.
2.	The model integrates multiple complex diffusion stages, which could complicate the training and debugging processes and make them difficult to optimize. Particularly, adjusting hyperparameters and verifying model stability might require additional effort.
3.	Given the complexity involved in the expectation state and k-step diffusion processes, the model may have high demands on computational resources.

**Questions:**

1.	How is the stability of subgraph selection ensured during the implementation of the multi-step subgraph diffusion (k-step same-subgraph diffusion) process? Is there a risk of accumulating long-term errors due to inappropriate subgraph choices?
2.	Does the training strategy for the subgraph prediction model $p_\partial(s_t|R^t)$ include handling of imbalanced data? Specifically, how does the model avoid bias towards subgraphs that appear more frequently in the training data?
3.	Can this method also be applied to broader graph representation learning tasks, such as node classification and clustering?
4.	Diffusing considering substructures is an interesting aspect from my perspective; besides the functional group substructures in molecular graphs, community substructures [1] are prevalent in other networks. If considering community structures during the diffusion process, what insights might the authors have? This is an interesting question, as community structures are widely present in biological and social networks.

[1] 'Community detection in graph: An embedding method' in IEEE TNSE

**Limitations:**

This paper introduces a complex diffusion model that leverages subgraph structures to enhance molecular representation learning. However, more limitations should be discussed, such as overfitting, computational efficiency, and generalization across diverse molecular structures.

---

> ### Author Rebuttal · Authors · 2024-08-07
>
> We thank the reviewer for the positive comments and thoughtful feedback on our work.
>
> **[W1. Highly specialized datasets]** Thanks for highlighting this important concern. The subgraph prediction model shares the molecular encoder with the denoising network and has an additional classification head. The subgraph predictor is trained together with the denoising network on the same dataset. We use the commonly used datasets GEOM and PCQM4Mv2 to ensure fair comparisons with the baselines. The overfitting risk can be avoided by using a larger dataset for pretraining. Our experiment in the cross-domain task shows that our proposed method has good generalization ability and may alleviate the overfitting problem to some extent.
>
> **[W2&W3. Multiple stages and complexity]** We appreciate the reviewer's concern about the complexity of SubgDiff. While we acknowledge our model is more complex than the vanilla diffusion model, we believe this complexity is acceptable and can be justified by performance improvements, especially in molecular force predictions. To address your concerns:
> - Training and debugging: compared to the original diffusion model, our model only adds a classification head for subgraph prediction and one additional loss. So the complication of training and debugging can be under control.
> - Optimization and hyperparameters: Our method brings two more hyperparameters: the weight $\lambda$ for subgraph prediction loss and $k$ for the $k$-step same-subgraph diffusion. We conduct ablation studies (see the Tables in the general response) and provide recommended hyperparameters based on our experiments.
> - Model stability: We implement gradient clipping, careful learning rate scheduling, and conduct long-term training experiments to ensure stability, which are the common techniques for diffusion model training.
> - Computational resources: In practice, compared with typical diffusion models, the introduced expectation state and $k$-step same-graph diffusion do not bring much computational overhead. We explain it from training and inference respectively.
> (1）Training: The expectation state and k-step diffusion only affect the weights of adding noise. SubgDiff can directly compute the $R^t$ from $R^0$ like the classical diffusion model as shown in Eq(16) of the main paper, i.e. $R^{t} = \sqrt{\frac{\bar\gamma_{t}\bar\alpha_{m}}{\bar\gamma_{km}}}R^0 + (\frac{\bar\gamma_{t}}{\bar\gamma_{km}} p^2 \sum_{l=1}^{m} \frac{\bar\alpha_{m}}{\bar\alpha_{l}}(1-\frac{\bar\beta_{kl}}{\bar\beta_{(l-1)k}}) + 1-\frac{\bar\gamma_{t}}{\bar\gamma_{km}})\epsilon$, where the weight calculations are scalar and do not require much computational overhead. In practice, the training time of one batch is similar to the GeoDiff baseline.
> (2) Inference: The sampling computation also only needs to calculate some scalar coefficients as shown in Eq（18). From Table 3, we can see that SubgDiff needs fewer sampling steps compared to baseline GeoDiff, demonstrating the computational efficiency of the proposed method.
> We also provide the source code [here](https://anonymous.4open.science/r/SubGDiff/README.md) to make SubgDiff accessible.
>
> **[Q1. Multi-step subgraph diffusion]** Thanks for the insightful question! During training, the subgraph will be randomly sampled from the subgraph set generated from the torsional-based decomposition method, which randomly selects a torsional bond among the given molecule and breaks the molecule into two parts. Each part is a connected subgraph and then randomly chosen as the subgraph. This predefined subgraph set will ensure the appropriateness of subgraph choices. For each time step, the probability of each node being included in the diffusion process is 0.5. We use a relatively large diffusion time step to make uniform diffusion for each node and ensure the avoid the risk of accumulating errors.
>
> **[Q2. Imbalanced data]** Thanks for the valuable question. The current training strategy does not consider this issue, so as to make a fair comparison with the baselines. In our subgraph sampling strategy, each node has the same probability of 0.5 to be included in the diffusion process. So as the diffusion step is large, our method is comparable to the baselines w.r.t the same training dataset. However, the imbalanced data is an important issue. It may be addressed by clustering the dataset and using a stratified sampling strategy during training.
>
> **[Q3. Broader graph representation learning tasks]** Thanks a lot for the insightful comments. In general, the introduced subgraph prediction can be used as an auxiliary loss for representation learning, if there is subgraph information available. According to the applications, our framework can be easily adjusted to incorporate substructure information into the representation learning model.
>
> **[Q4. Community substructures]**
> Thanks for raising this interesting question regarding the potential application of the substructure-aware diffusion process to community structures in other networks. We're excited to share our thoughts on this matter:
> - The parallel between functional group substructures in molecular graphs and community structures in other networks is indeed compelling. This analogy suggests that our approach could potentially be adapted to other domains with hierarchical or modular structures.
> - Applying a similar diffusion process to networks with community structures could potentially yield several insights, such as community evolution over time, inter-community interactions, and hierarchical community structure.
> - There exist potential applications in other domains such as biological networks or social networks. In biological networks, a community-aware diffusion model could potentially: a) Predict the impact of perturbations (e.g., gene knockouts) on functional modules; b) Identify key regulatory hubs that influence multiple communities.
>
> **[Limitataions]**
> Thanks so much for the advice. We will add those discussions to the revised paper.

---

> > ### Comment · Reviewer_Djew · 2024-08-11
> >
> > Thank you to the authors for their reply, which has resolved most of my issues. I will maintain my positive rating, and I wish you good luck.

---

> > > ### Author Response · Authors · 2024-08-12
> > >
> > > Dear reviewer Djew,
> > >
> > > Thank you so much for the positive rating!
> > > We sincerely appreciate your constructive suggestions and valuable comments for improving our paper. Thank you!
> > >
> > > Best regards,
> > > The Authors

---

### Official Review · Reviewer_BpDA · 2024-07-19

**Soundness:** 2
**Presentation:** 3
**Contribution:** 2
**Rating:** 5
**Confidence:** 3

**Summary:**

The paper proposes a diffusion-based pretraining method using subgraphs to learn enhanced molecular representations. Unlike previous methods which normally add noise to every atom, this paper proposes adding noise based on subgraphs. The method is evaluated on various downstream tasks to demonstrate its effectiveness, such as 2D and 3D property prediction tasks.

**Strengths:**

- The paper offers an interesting perspective on existing molecular diffusion models and proposes several approaches to address them accordingly, which is inspiring.

- The paper presents many experiments and analysis to illustrate the results, which could provide valuable insights to the community.

**Weaknesses:**

- Although the proposed method effectively addresses the identified limitations, it does not provide much chemical intuition for the design. The authors claim that existing methods neglect the dependency in substructures, but the proposed method does not seem to integrate or learn the inherent molecular substructure information. The decomposition does not seem to be based on significant chemical knowledge, and the interactions or relations between various substructures are not explored. The motivation is not heavily grounded in domain knowledge.

- The method is a little bit confusing. The entire training process includes many steps and three key training objectives: subgraph prediction, expectation state, and k-step same-subgraph diffusion. The paper mainly describes each component separately, but it is unclear how the three objectives are leveraged during the training. Some details are also not clear. For example, “The mask vector $s_t$ is sampled from a discrete distribution $p\_{s_t}$ (S|G)”—does $p_{s_t}$ vary for each molecule? How exactly is the distribution obtained? Also, an overall framework could help better understand the process.

- The diffusion steps number is 5000, which is quite large. Molecular property prediction datasets normally contain small molecules. Are there any specific reasons for using such large steps? What are the computational costs?

- The proposed method seems impractical due to its complexity, while the performance improvement is not very significant. It is difficult to justify the trade-off between model complexity and performance. Perhaps the authors could provide a more in-depth discussion to advocate for their method compared to other baselines, beyond just the prediction performance.

- Some ablation studies are not conducted. For example, k seems to be an important h-param. What are the effects of different k values? Also, what about the performance of applying only one training objective?

**Questions:**

See weaknesses.

**Limitations:**

Discussed.

---

> ### Author Rebuttal · Authors · 2024-08-07
>
> We thank the reviewer for the valuable comments and positive feedback!
>
> **[Weakness 1. Chemical intuition]**
> Thank you for your insightful comment! We agree that chemical intuition and domain knowledge are crucial in molecular representation learning, and we'd like to address how our method incorporates these elements:
> (1) While our method may not explicitly learn predefined substructures, it does capture atomic dependencies within the substructure through diffusion on subgraphs. This approach allows the model to implicitly learn relevant substructure information from the data, rather than relying on predetermined chemical fragments.
> (2) Further, we sincerely clarify that the subgraph is obtained by decomposing the molecule by breaking the rotational bonds. This torsional-based decomposition method can capture the inherent molecular 3D substructure information. For example, the benzene ring is always preserved in the same subgraph since the π bonds in benzene rings aren't torsional. The downstream tasks on 3D property prediction also demonstrate the effectiveness of our method.
>
>
> **[Weakness 2. Overall methods]**
> We thank the reviewer for this valuable feedback. We acknowledge that our method involves multiple components, which may have led to some confusion. We apologize for any lack of clarity in our presentation and would like to provide a more comprehensive explanation of how these components work together:
> (1) As we described in section 4.5, in the forward process, we use **expectation state diffusion** to get the state $km$ ($m:=\lfloor t/k \rfloor$) from the initial state $0$ (Phase I). Then the $(t-km)$-step **same-subgraph diffusion** is employed to get the state $t$ from state $km$ with the same subgraph $s_{km+1}$ (Phase II). In the training step, we use the denoising loss and the **subgraph prediction** loss to train the model (Equation. 16).
> (2) These components are not independent but rather complement each other:
> - The subgraph prediction task helps the model learn local structural features.
> - The expectation state reduces the complexity of $R^t$ and removes the unstable sampling of the masks $s_{1:km}$.
> - The k-step same-subgraph diffusion accumulates more noise on the same subgraph from $km$ to $t$ for facilitating the convergence of the subgraph prediction loss.
>
> (3) The subgraph is represented by a mask vector $s_t$. The distribution $p_{s_t}(s|G)$ is implicitly defined by the torsional-based decomposition method. One can get two subgraphs by breaking a torsional bond. Thus there are $2n$ subgraphs if the molecule has $n$ torsional bonds. $p_{s_t}(s|G)$ is a random distribution over the $2n$ subgraphs. And it varies for each molecule.
>
> **[Weakness 3. Diffusion steps number]**
> Thank you for bringing up this important point about the number of diffusion steps and computational costs. From our experiments, we find that using 5000 steps can obtain better performances, as shown in Table 3 for conformation generation. We also evaluate the downstream 3D property prediction tasks when pretraining with different steps, and the results are shown in **Table #3** of the general response.
> The large diffusion step will not increase the computational cost for the downstream tasks, as we use the diffusion denoising loss for pertaining. We don't need to do the sampling process that requires high computational overhead when we predict the molecular property.
>
> **[Weakness 4. Complexity]**
> Thank you for bringing this important concern to our attention.
> (1) As we have shown in the paper, our method significantly outperforms baselines on 3D molecular property prediction, as shown in Table 1 in the main text and Table 8 in the appendix. In addition to the prediction performance, our method also brings the sampling efficiency and generalization ability. The results from Table 3 show that SubgDiff significantly outperforms the baseline when adopting fewer diffusion steps. As shown in Table 4, SubgDiff consistently outperforms the other baselines in the cross-domain conformation generation tasks.
> The above discussion will be added to the final version.
> (2) Despite the multiple components of the algorithm, the method is practical to implement. We train the model using the objective in Eq(17), which is similar to the typical diffusion model DDPM. The main difference with DDPM are the noisy version $R^t$ computed from Eq(16), i.e. $R^{t} = \sqrt{\frac{\bar\gamma_{t}\bar\alpha_{m}}{\bar\gamma_{km}}}R^0 + (\frac{\bar\gamma_{t}}{\bar\gamma_{km}} p^2 \sum_{l=1}^{m} \frac{\bar\alpha_{m}}{\bar\alpha_{l}}(1-\frac{\bar\beta_{kl}}{\bar\beta_{(l-1)k}}) + 1-\frac{\bar\gamma_{t}}{\bar\gamma_{km}})\epsilon$, where the weighting coefficients can be obtained simply by scalar calculation. This brings very little complexity. In practice, the training speed is similar to the original diffusion model (GeoDiff). In addition, we also provide the source code [here](https://anonymous.4open.science/r/SubGDiff/README.md), which will make it easy to use our method.
>
> **[Weakness 5. Ablation]** Thank you for the constructive suggestion. (1) We provide the sensitivity analysis of $k$ in $k$-step same-subgraph diffusion. The results are shown in **Table #1** of the general response.
> (2) When only using the denoising loss as the training objective, without subgraph prediction loss, the method cannot be used for sampling due to the inaccessible $s_t$ during inference. Nevertheless, the pure denoising objective can be used for pertaining and the results of the downstream force prediction tasks are shown in **Table #2** of the general response.

---

> > ### Author Response · Authors · 2024-08-12
> > **Kind reminder to the reviewer**
> >
> > Dear reviewer,
> >
> > We kindly ask if you could inform us whether your concerns have been adequately addressed in our rebuttal, or if you have any further questions. We are committed to responding promptly to any additional inquiries you may have.
> >
> > Thank you for your time and valuable feedback.
> >
> > Best regards,
> >
> > The Authors

---

> > ### Comment · Reviewer_BpDA · 2024-08-13
> >
> > Thank the authors for the responses, which seem sound to me. I am maintaining my score since I am not an expert in diffusion models. I suggest that the authors include these additional information in the revision to make the paper more clear.

---

### Author Rebuttal · Authors · 2024-08-07

## General Response

Dear reviewers,

Thanks to all the reviewers for your time and effort during the review process and the constructive advice. In addition to the response to each reviewer individually, we conduct the ablation study and sensitivity analysis of the k-step same subgraph and diffusion steps for our method. The results are shown below in Table #1, Table #2, and Table #3.
We also provide curves of subgraph prediction error during training in the attached PDF.

**Table #1. The sensitivity analysis for different $k$ in $k$-step same subgraph diffusion on conformation generation.**
| k | 1(ld sampling*) | 10 | 25| 50 |
|----------|---------|----------|----------|--------------|
|  COV-R(Mean)     ↑    |     89.70    |     88.06             |     **89.78**     |       89.02    |
|   COV-R(Median) ↑       |  93.96      |      93.26      |  **94.17**         |    93.21      |
|   MAT-R(Mean) (Å)   ↓     |     0.5235     |     0.2623     |      **0.2417**          |      0.2706    |
|   MAT-R(Median) (Å)   ↓     |    0.2710      |      0.2597    |       **0.2449**       |       0.2709    |
|  COV-P(Mean)  (%)  ↑      |      49.90    |       47.38     |         **50.03**      |          48.63|
|  COV-P(Median)  (%)  ↑      |   47.00       |      47.06    |           **48.31**        |          46.77|
| MAT-P(Mean) (Å) ↓ |      4.7816    |        4.2922 |                   **0.5571**              |  0.7512       |
| MAT-P(Median) (Å) ↓ |   0.5378       |   0.5615       |             **0.4921**          |    0.4995      |

*The subgraph predictor cannot predict the correct subgraphs so we use ld sampling rather than DDPM when k=1.

**Table #2 Ablation study for the pretrained model and evaluate the model on MD17 force prediction (MAE) downstream task.**
| Component | Aspirin ↓ | Benzene ↓ | Ethanol ↓ | Malonaldehyde ↓ | Naphthalene ↓ | Salicylic ↓ | Toluene ↓ | Uracil ↓ |
|--------------------|-----------|-----------|-----------|-----------------|---------------|-------------|-----------|----------|
| w/o subgraph prediction loss| 1.193     | 0.305     | 0.321    | 0. 478  | 0.456 | 0.678       | 0.406     | 0.470    |
| w/o k-step same subgraph| 1.011     | 0.278     | 0.289    | 0. 461  | 0.448 | 0.665      | 0.377     | 0.464    |
| w/o expectation state| 0.931     | 0.281     | 0.276    | 0. 465  | 0.421 | 0.601       | 0.380     | 0.446    |
| expectation state + subgraph prediction loss | **0.880** |**0.252** | **0.258** | **0.459**   | **0.325** | **0.572**   | **0.362** | **0.420**|

**Table #3 The sensitivity analysis of diffusion steps for the pretrained model and evaluate the model on MD17 force prediction (MAE) downstream task.**
| #diffusion steps| Aspirin ↓ | Benzene ↓ | Ethanol ↓ | Malonaldehyde ↓ | Naphthalene ↓ | Salicylic ↓ | Toluene ↓ | Uracil ↓ |
|--|--|-|-|--|-|-|-|-|
| 100 | 0.921     | 0.282       | 0.275     | 0.471    | 0.348 | 0.596   | 0.394     | 0.465    |
| 1000 | 0.901     | 0.261     | 0.266    | 0. 463  | 0.336 | 0.590       | 0.385     | 0.438    |
| 5000    | **0.880** |**0.252** | **0.258** | **0.459**       | **0.325**     | **0.572**   | **0.362** | **0.420**|


Best regards,

The authors

---

### Decision · Program_Chairs · 2024-09-25

**Decision:**

Accept (poster)

**Comment:**

This paper proposes a subgraph-based diffusion model for molecular representation learning, where noises are added to subgraphs in the diffusion process. Experimental results in multiple different tasks including molecular representation learning and conformation generation prove the competitiveness of the proposed approach over existing baselines. Overall, the subgraph-based diffusion model is new and novel. The experiments are strong.